# Latent transition analysis for longitudinal studies of post-acute infection syndromes

Roy Gusinow [1,2], Anna Górska [3], Lorenzo Maria Canziani [3], Iris Lopes-Rafegas [4,5], Carolina Alvarez Garavito [1,2], Adriana Tami [6], Elisa Gentilotti[3], Elisa Sicuri [4,5,7,8], Cédric Laouénan[9,10], Jade Ghosn [9,10], Aline-Marie Florence[9,10], Nadhem Lahfej[11], Fulvia Mazzaferri [3], Lidia Del Piccolo [12], Maddalena Giannella[13,14], Alice Toschi[14], Michela Di Chiara [14], Maria Giulia Caponcello[15,16], Zaira R. Palacios-Baena[15,16], Karin I. Wold [6], Elisa Rossi[17], Evelina Tacconelli[3,32] ✉ & Jan Hasenauer [1,2,32] ✉
On behalf of the ORCHESTRA study group*

Post-Acute Infectious Syndromes (PAIS) refer to the symptoms persisting months after initial infection. Clinical research studies on this topic often collect rich, multi-modal datasets. Yet, the complexity of the datasets and the lack of a precise clinical case definition pose difficulties in creating comprehensive analyses. Here, we present a generalisable framework for analysing data from longitudinal studies of PAIS using Latent Transition Analysis (LTA). It enables the identification of disease phenotypes and the patient-level analysis of transitions between them, without relying on predefined clinical categorisations. Furthermore, we introduce a method for incorporating covariate information, which enables exploration of how patient characteristics influence disease trajectories. We apply this methodology to the ORCHESTRA dataset, composed of individuals affected by SARS-CoV-2 infection from multiple European centres, for investigation into Post-COVID-19 condition (PCC). 5094 patient assessments were collected at SARS-CoV-2 infection, and at 6, 12, 18, and 24 months of follow-up. Our model identifies distinct PCC phenotypes with patient trajectories impacted by age and sex. Our results highlight how LTA can enhance the interpretability of complex, time-resolved clinical data, support personalized patient monitoring and management, and accelerate therapeutic development for other PAISs, too.

Understanding and managing the long-term health consequences of acute diseases is a critical component of public health. A number of infectious pathogens can cause Post-Acute Infection Syndromes (PAIS), a source of unexplained chronic disability with diverse and often severe symptoms lasting for months after the resolution of the initial acute infection[1]. Longitudinal studies of PAIS are instrumental in identifying lasting effects, tracking symptom trajectories, and guiding interventions. These syndromes present substantial challenges to both research and clinical care due to their protracted nature, heterogeneity, and often unclear pathophysiology.

One impactful example of PAIS is Post-COVID Condition (PCC), also known as Long COVID or Post-COVID-19 Syndrome. According to the World Health Organization, PCC is defined as the presence of symptoms lasting more than two months following a confirmed or probable SARS-CoV-2 infection, without an alternative diagnosis[2]. It is

---

A full list of affiliations appears at the end of the paper. *A list of authors and their affiliations appears at the end of the paper.
✉e-mail: evelina.tacconelli@univr.it; jan.hasenauer@uni-bonn.de

estimated that more than 65 million people globally are affected by PCC[3], making it a major health and societal burden.

Current efforts to delineate PCC phenotypes rely on symptom clustering techniques, typically applied to cross-sectional data. These studies have consistently identified symptom-based phenotypes such as Chronic Fatigue, Respiratory, Pain, Neurosensorial, and Gastro-intestinal clusters[4,5]. A complementary line of research has employed severity-based clustering, grouping patients by symptom burden rather than type[6,7]. Incorporation of Health-Related Quality of Life (HRQoL) measures has further highlighted the substantial impact of PCC on daily functioning[8,9]. Risk factor analyses suggest higher susceptibility among women, unvaccinated individuals, patients with diabetes or chronic respiratory disease, and those with severe acute illness[10].

However, these clustering approaches face important limitations. Clusters are often defined independently at each timepoint as the condition can present differently in patients from one timepoint to the next, making it difficult to track the progression of disease states or to compare clusters across assessments. Symptom profiles can fluctuate, resulting in false negative observations, variable cluster composition, and inconsistent phenotype definitions[11]. While unsupervised learning methods such as hierarchical clustering help address some of this subjectivity[12,13], they still fail to capture the temporal dynamics of patient trajectories or the influence of individual-level characteristics over time.

To overcome these limitations, researchers have turned to Latent Transition Analysis (LTA) – a type of Hidden Markov Model (HMM) designed to infer unobservable (latent) states and estimate transition probabilities between any two states over successive timepoints[14]. LTA is particularly well-suited for studying PAISs, where symptom presentation is presumptively dynamic, and the underlying disease structure is not directly observable.

LTA has already been applied to PCC in several recent studies. It was used to investigate trajectories of quality-of-life impairments[15], explore psychological distress patterns[16], and study the evolution of pulmonary function after infection[17]. These efforts demonstrated the utility of LTA, yet they were constrained by modest cohort sizes (largest: 1467 patients) and the number of latent states (5 states), limited observational modalities (only binary or continuous data), and restricted model parameterisations (number of covariates), reducing their capacity to generalize findings or support individual-level predictions.

In this study, we demonstrate how LTA, capable of considering a large number of latent states and a set of covariates estimated in an efficient manner, can be used for the extensive modelling and model-based assessment of data from PAIS studies. The flexible framework enables the data-driven identification of latent disease phenotypes, the modelling of temporal disease progression, and the analysis of covariate-dependent transition dynamics, without relying on pre-defined clinical categorisations. To support personalised patient-level insights, we further describe a filtering-based approach that improves the prediction accuracy of individual symptom trajectories. While the methodology is broadly applicable to a wide range of chronic and post-infectious conditions, we showcase it by studying PCC using a partially published dataset collected within the ORCHESTRA project. This rich, multinational cohort includes longitudinal clinical, symptom, and quality-of-life data from over 5000 microbiologically confirmed SARS-CoV-2 patients across four countries, with follow-up extending up to 24 months.

The LTA framework is tailored for heterogeneous longitudinal data, with the incorporation of demographic and clinical covariates into the model's transition structure. It allows for the identification of robust PCC phenotypes and transition pathways, as well as the interpretation of key risk and protective factors governing disease progression and recovery. Our approach provides a transferable template for investigating other PAISs, offering both mechanistic insight into the hidden patient health progressions and practical utility of risk and protective factors for clinical monitoring and therapeutic development.

## Results

### A flexible latent transition analysis framework for long-term Sequelae studies

To provide a comprehensive and scalable assessment of PAIS, we propose a flexible and interpretable LTA framework (see Fig. 1 for a visual outline). The framework builds on established Hidden Markov Modelling implementations[18,19], addressing the specific challenges of longitudinal cohort studies with heterogeneous data types (i.e., binary and continuous), as well as incomplete observations that arise over long follow-up periods.

In order to circumvent the large number of parameters present when modelling the full matrix of covariate effects separately for each transition probability, patient characteristics are projected onto a low-dimensional scalar representation that modulates the entire transition matrix in an interpretable and computationally efficient way.

We conducted a comprehensive simulation-based validation study to evaluate the robustness of parameter recovery, predictive performance, and covariate interpretability across a range of controlled scenarios. These experiments confirm the framework's reliability under varying data sparsity, symptom noise, and covariate effects, providing confidence in its application to real-world longitudinal data. Indeed, we found our approach to significantly reduce overfitting risk while preserving individual-level heterogeneity.

In addition to characterising state transitions, our framework supports state filtering for individual patients, making use of prior symptom and HRQoL history to improve future symptom and HRQoL predictions (Fig. 1f). By applying a recursive state update procedure, we dynamically refine patient-level latent state probabilities at each timepoint and generate predictions for both binary and continuous variables.

Full methodological details, including model specification, estimation procedures, and evaluation metrics, are provided in the Methods section. The full implementation of the framework is provided, including model fitting, prediction, and visualisation routines, available on https://doi.org/10.5281/zenodo.17787061.

### Clinical characterisation of the ORCHESTRA Long-Term Sequelae Cohort

In order to assess the proposed framework for LTA in PAIS, we analysed data from the ORCHESTRA Long-Term Sequelae Cohort. This cohort of individuals with confirmed SARS-CoV-2 infection was previously established as part of the EU-funded ORCHESTRA project[20] and comprises six prospective subcohorts from 56 centres across five countries (Fig. 2a). Cohort included adult patients (>14 years old) with laboratory-confirmed SARS-CoV-2 infection. At baseline, data collection included demographic characteristics, comorbidities, clinical severity, ICU admission, early antiviral and monoclonal antibody treatment (Fig. 2b–e, Supplementary Data 1). HRQoL measures through the Short Form Health Survey 36 (SF-36) were collected during each follow-up. A poor HRQoL was defined for a score below 50 (Fig. 2f, Supplementary Data 3). In addition, we focus on 9 PCC-related symptoms (ageusia, anosmia, arthralgia, cough, dyspnoea, fatigue, headache, memory loss and myalgia) measured at all timepoints (Fig. 2g and Supplementary Data 2).

In total, the dataset contained observations from 5094 individuals, all of whom had data at the acute phase and at least one follow-up (Supplementary Data 1). The dataset includes 1796 patients for whom analysis of the 12-month follow-up data was published before[21], while data from the 18- and 24-month visits are presented here for the first time. The assessment of the newly collected data revealed a minor drop in patient participation from the 12- to 18-month follow-up (from 2495 to 2120), while 628 patients completed a 24-month visit

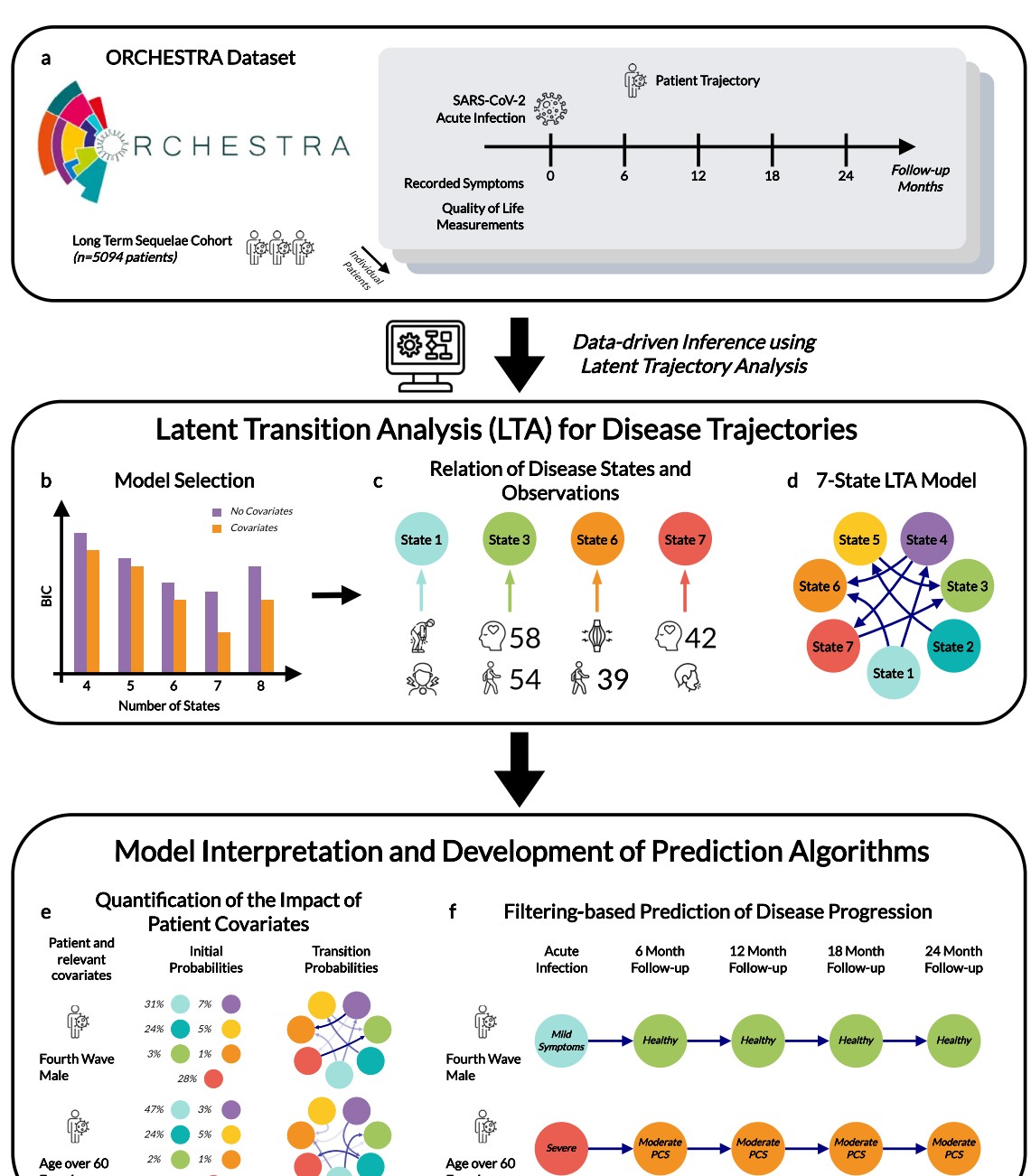

**Fig. 1 | Illustration of Latent Trajectory Analysis for Longitudinal Studies of Post-Acute Infection Syndromes, using the PCC ORCHESTRA Study as an Example. a** Data collection process which provides information about the health state for each patient (recorded symptoms and HRQoL measurements) at different timepoints. **b** Parameterisation of model candidates and model selection. **c** Interpretation of model states (using symptoms and HRQoL scores) and **d** relationship between states for chosen 7-State Model. **e** Model-based quantification of the impact of patient covariates and their relation to observed symptoms. Percentages next to the coloured circles represent the conditional probabilities of the identified states given patient covariates. **f** Prediction of trajectories of health states from the acute infection. As new observations are recorded throughout, trajectories are updated.

(Supplementary Fig. 1). A total of 419 patients were assessed at all five timepoints, with most participants having assessments at both acute infection and the first follow-up at 6 months. Notably, there is a non-negligible number of patients with non-standard follow-up patterns, including a subgroup of 274 individuals assessed at 18 months who had data only from the acute stage prior to that assessment time period.

### Data-informed latent transition analysis identifies initial COVID-19 infection and post-COVID syndrome states

To comprehensively characterise the disease dynamics and individual symptom trajectories associated with PCC based on the ORCHESTRA Long-Term Sequelae Cohort, we applied our LTA framework, using a fully connected HMM allowing for transitions between all pairs of latent health states, without imposing any a priori structure on the symptom profiles or progression paths (Fig. 3). Each latent state is defined by a set of emission probabilities for binary symptoms and expected values for continuous HRQoL scores, enabling joint modelling of discrete symptom and continuous HRQoL data. The model also captures patient-level heterogeneity through covariate-dependent transition dynamics, with state identities and trajectories inferred directly from the data via maximum likelihood estimation.

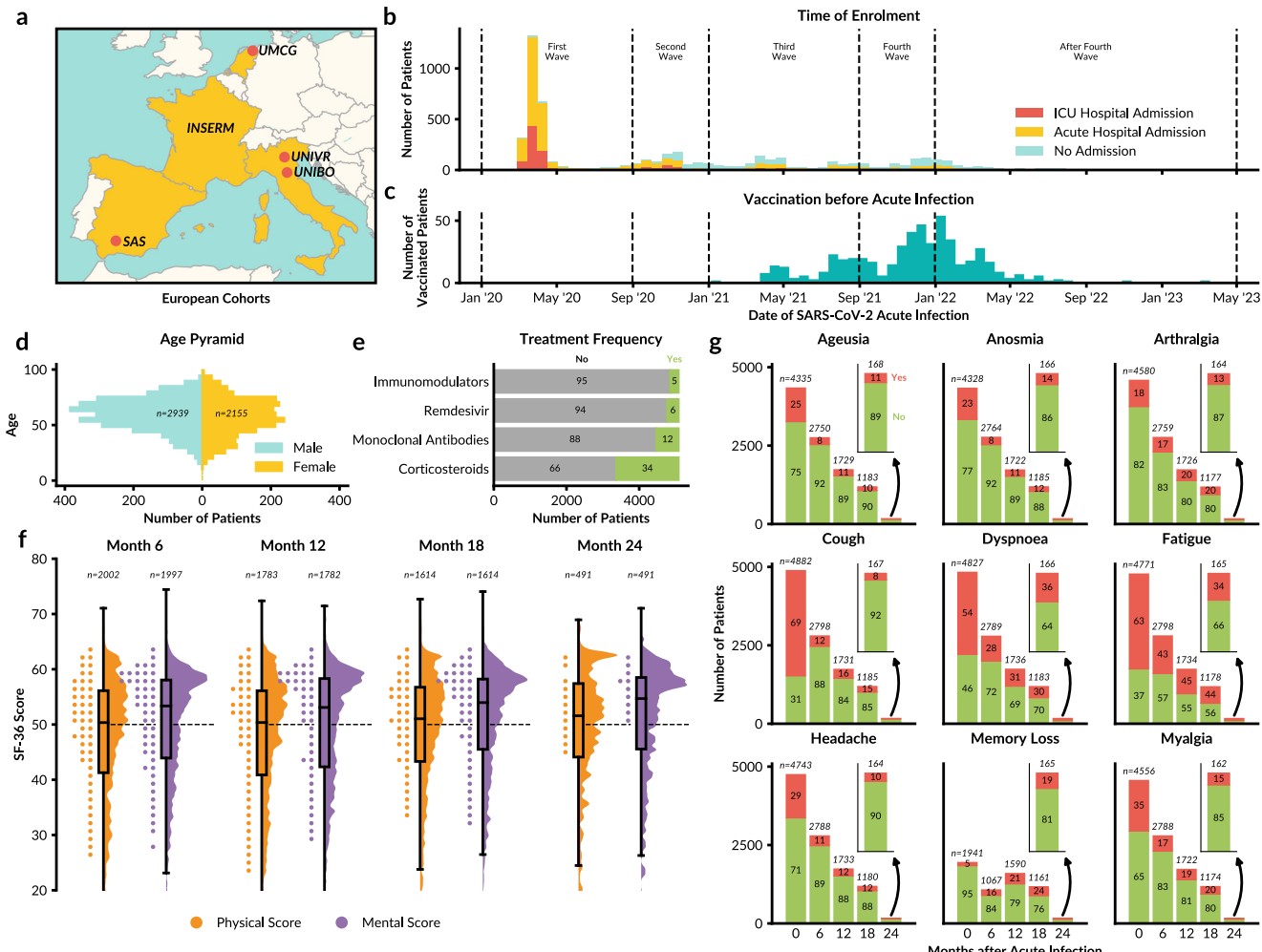

**Fig. 2 | ORCHESTRA Long-Term Sequelae Cohort Dataset. a** Prospective cohorts included in ORCHESTRA study; Servicio Andaluz de Salud (SAS), Institut national de la santé et de la recherche médicale (INSERM), University Medical Center Groningen (UMCG), University of Verona (UNIVR), and University of Bologna (UNIBO). **b** Stacked SARS-CoV-2 infection timeline of patients, separated by level of care. **c** Number of enrolled patients vaccinated for SARS-CoV-2 before acute infection over time. **d** Age and sex distributions. **e** Distribution of COVID-19 pharmacological treatment, including immunomodulators (i.e., tocilizumab, ruxolitinib, adalimumab, baricitinib, or tacrolimus) and anti-SARS-CoV-2 spike monoclonal antibodies (i.e., bamlanivimab, bamlanivimab/etesevimab, and casirivimab/imdevimab), measured at the acute phase. **f** SF-36 mental and physical component scores across the recorded months after acute infection. Box plots show the median (centre), first (Q1) and third (Q3) quartiles (box bounds), as well as Q1/Q3 - 1.5 × interquartile range indicated by the whiskers. The black dashed line at 50 indicates the norm-based scoring mean of the 1998 US general population (mean = 50, SD = 10). **g** Stacked bar charts of the 9 most commonly reported PCC-related symptoms over time. The total number of available patients is shown at the top of each bar, with the percent occurrence of symptoms shown within each bar. In total, the dataset contained observations for 5094 individuals (Supplementary Data 1).

By inspecting the emission probabilities, states were interpreted as representing distinct health profiles, where the labelling process involved close examination of the emission patterns by clinical experts to identify the clinical meaning of the state. For example, a state with very low probabilities of manifesting any symptom and high probability of high HRQoL scores can be clearly annotated as a healthy state.

The assessment of the models with 4 to 8 latent states revealed a high degree of structural similarity (Supplementary Fig. 3). Model selection reveals that the 7-state model is most appropriate (Supplementary Fig. 2). This model provided clinical interpretations for each state. States 1–2 were exclusively observed during the acute infection in the acute phase, while States 3-7 were predominantly observed in follow-up timepoints (Fig. 3a).

At the extremes, we identified the Healthy state (State 3) and the Severe Symptom state (State 7) based on emission probabilities (Fig. 3b). The Healthy state is characterised by very low symptom probabilities (<15% for fatigue, <10% for all other symptoms), indicating complete recovery from the initial COVID-19 infection. This interpretation is further supported by notably high mean SF-36 scores for

both physical (54.31) and mental (57.62) components, reflecting above-average HRQoL. In contrast, the Severe Symptom state displays consistently high probabilities across all symptoms and notably lower SF-36 scores, indicative of significantly compromised HRQoL.

In addition to these extreme states, the LTA reveals the Sensorial PCC state (State 4), the Fatigue PCC state (State 6), and the Respiratory PCC state (State 5) (Fig. 3b). The Sensorial PCC state is specifically characterised by elevated probabilities of anosmia (81%) and ageusia (89%), second only to the Severe Symptom state. Notably, this Sensorial PCC state predominantly impacts physical HRQoL but has minimal influence on mental health scores. Fatigue PCC is marked by elevated probabilities of fatigue (85%), along with intermediate probabilities (30−60%) for all other symptoms except anosmia and ageusia. The Respiratory PCC state shows slightly reduced physical and mental HRQoL scores (46.86 and 47.72, respectively) compared to the Healthy state, with notable increases in symptom probabilities for fatigue (from 6% to 31%) and dyspnoea (from 14% to 43%).

The analysis of the mean initial state distribution and transition dynamics reveals that two states exclusively emerge during the acute

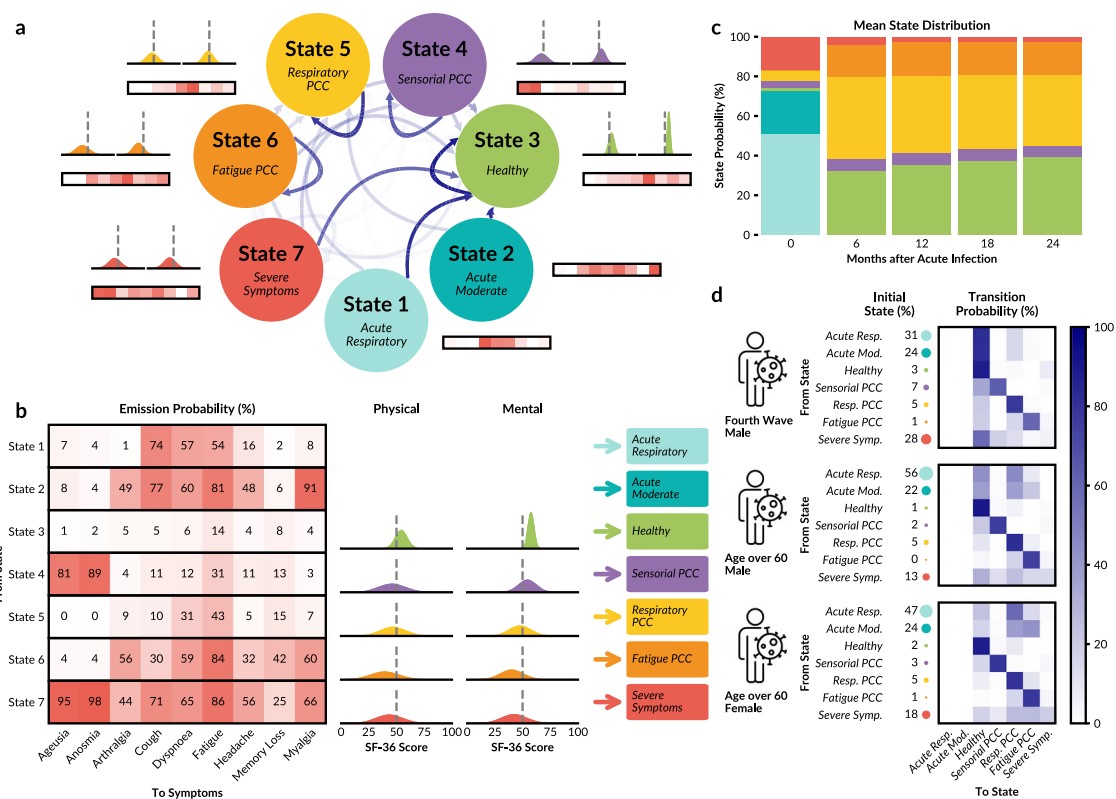

**Fig. 3 | LTA model for ORCHESTRA cohort. a** Model with 7 latent states: the intensity of the arrow colour indicates the average transition probability over the patients in the cohort. Each state has an associated emission probability for each of the nine PCC-related symptoms, as well as Gaussian distributions describing physical (left) and mental (right) HRQoL scores. **b** Heatmap showing the probability of reporting one of the 9 PCC-related symptoms given the state with its associated Gaussian distributions of the SF-36 HRQoL physical and mental component scores. **c** Mean probability of state at each timepoint. **d** Initial and transition state probabilities for three example patients: men infected during the fourth COVID-19 wave, men older than 60, and women over 60.

infection phase (Fig. 3c). The Acute Respiratory state (State 1) and Acute Moderate state (State 2) are characterised by high probabilities of cough (74% and 77%, respectively), dyspnoea (57% and 60%), fatigue (54% and 81%), and headache (16% and 48%), with Acute Moderate additionally characterised by arthralgia (49%) and myalgia (91%). The HRQoL distributions associated with both states have extremely large uncertainty in their means and standard deviations, which is expected, given that no HRQoL data were collected at the acute timepoint (Supplementary Tables 2 and 3). From 6 months onwards, the Healthy and Respiratory PCC phenotypes become dominant, with Sensorial PCC, Fatigue PCC, and Severe Symptom phenotypes having lower but non-negligible probabilities (6.0%, 17.0%, and 2.3%, respectively, at 24 months). Generally, patients tend to enter the Healthy state, indicating recovery, as time progresses (from 32.6% at 6 months to 39.2% at 24 months), mainly due to decreases in the probability of the Respiratory PCC state (from 41.5% at 6 months to 35.5% at 24 months) (Supplementary Table 6).

LTA reveals that patient characteristics influence initial state and state transitions in distinct ways. Across three patient profiles – chosen to cover the spectrum of disease presentation – the Acute Respiratory and Acute Moderate states, as well as the Severe Symptom state, are notably prevalent at the initial infection stage. This highlights the considerable number of patients experiencing extensive symptoms (Fig. 3d). The transition probabilities indicate that patients have a high likelihood of remaining within the Healthy, Respiratory PCC, Sensorial PCC, or Fatigue PCC states, suggesting that patients who transition into these states generally remain stable over subsequent follow-up periods. Given the initial prominence of the Acute Respiratory, Acute Moderate, and Severe Symptom states, the likelihood of transitioning out of these states during follow-ups is particularly important as

subsequent states typically exhibit high persistence. For instance, men infected during the fourth wave show a large probability of rapidly transitioning to the Healthy state, despite initially having a higher probability of starting in the Severe Symptom state (28%). Conversely, women older than 60 years show higher initial probabilities for the Acute Respiratory and Acute Moderate states (47% and 24%, respectively) and tend to move into Respiratory PCC and Fatigue PCC states during later follow-ups. These patients also exhibit higher probabilities of remaining within these PCC states, suggesting a comparatively lower chance of recovery.

## Cohort Level Dynamics Captured across Follow-up Symptoms and Health-Related Quality of Life Metrics

In order to assess the model's ability to accurately describe the observed cohort data, we conducted forward simulation using the patient characteristics of the ORCHESTRA dataset (see Forward Simulations). Since the true state of each patient is unobserved, our evaluation focuses on the distribution of symptoms, the HRQoL values, and their correlation structure.

The comparison of the model simulations with real observed data for the nine reported symptoms reveals an overall good agreement. For most symptoms and timepoints, the range of model simulations overlaps with 2 × standard error of the mean derived from the observed data (Fig. 4a). A discrepancy occurs in the predicted prevalence of dyspnoea at 24 months, where simulations typically forecast a decrease, while the observed data indicate an increase. This anomaly could reflect potential sampling bias at the 24-month follow-up, which is also evident from the broader confidence intervals. Similarly, the memory loss symptom exhibits a minor increase from 12 to 18 months but a slight decrease at 24-months, suggesting potential

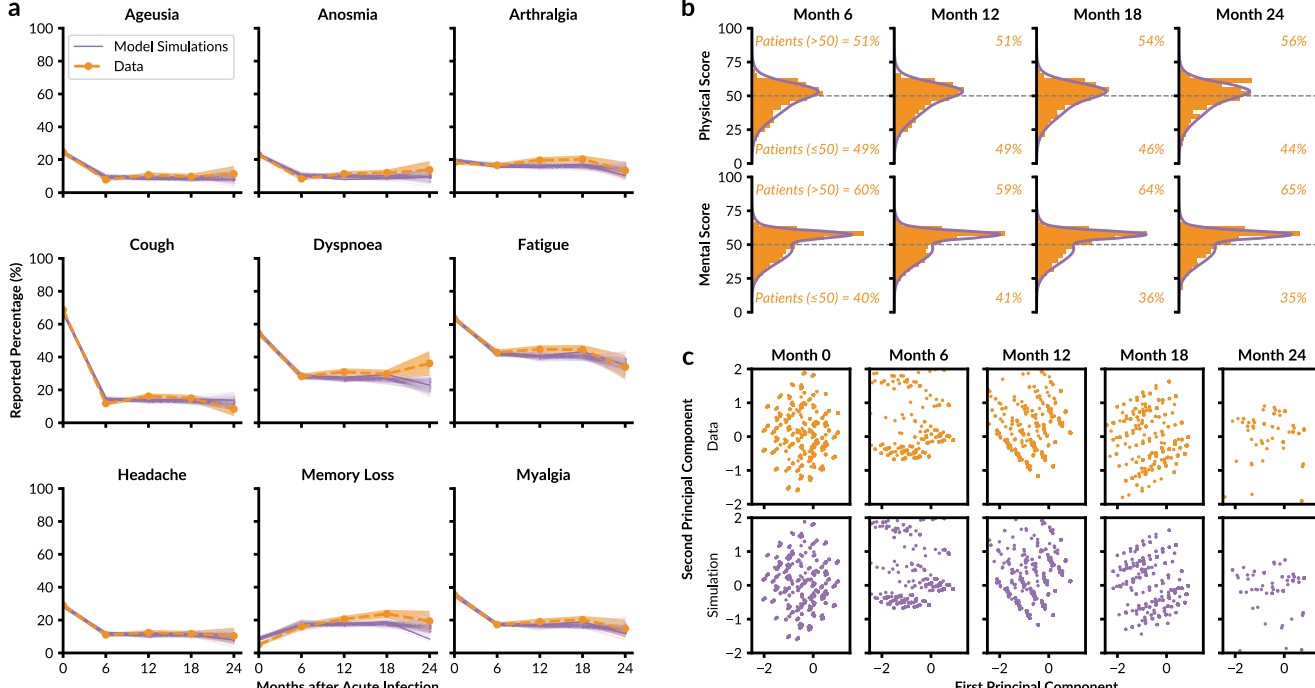

**Fig. 4 | Comparison of model and data at the cohort level. a** Percentage of the cohort reporting a specific symptom at Months 0 to 24. The observed dataset is indicated in orange (bold dashed line: mean; shaded area: 2× standard error of the mean) and the results of 1000 model simulations are indicated in purple (bold line: mean; thin lines: individual simulations). **b** Reported SF-36 physical and mental component scores (orange) and distribution from model simulations (purple) for timepoints after acute infection. **c** Dimension reduction for the vectors containing symptom information and SF-36 physical and mental component scores. The reported datapoints (orange) and simulation results by the model (purple) are indicated in the space of the first two principal components of each patient.

fluctuations or biases in symptom reporting over extended follow-up periods.

The assessment of the distribution of the HRQoL scores revealed strong agreement between averaged model simulations and observed data across all timepoints (Fig. 4b). Despite the complex structure of the mental health scores, characterised by a peak around 60 and a long tail extending toward lower values, the simulations are able to accurately capture this dynamic. Indeed, the model predicts that distinct segments of these distributions correspond to specific latent states (Supplementary Fig. 5). The pronounced clustering around 60, accompanied by a secondary cluster around 45, is driven primarily by patients transitioning into Healthy and Respiratory PCC states. Conversely, patients classified within the Respiratory PCC state commonly report scores around 50 for both the mental and physical components, indicating a stable but mildly impacted HRQoL. Over time, patients do recover in both physical (51% of patients with a score above 50 at Month 6 to 56% at Month 24) and mental components (60% of patients >50 at Month 6 to 65% at Month 24) of the SF-36.

Inspection of the Principal Component Analysis (PCA) plots of the cohort data shows that the projected datapoints are not easily separable between timepoints, implying that it is difficult to discern underlying clusters (Fig. 4c). Yet, the LTA provides a model with symptom observations that map to the first two principal components. These are similar in structure to the real observation, displaying the ability to capture underlying latent mechanisms which would normally not be revealed by a standard factor analysis[12,22,23]. More so, the PCA loading values of the first principal component also match closely between the dataset and simulations, implying the impact of symptoms on the underlying patterns is also similar (Supplementary Fig. 6). In addition, comparisons of the Pearson correlation coefficients between symptoms and HRQoL variables produced by the dataset and averaged simulation also yielded similar results (Supplementary Fig. 7).

Our assessment of the model at the cohort level revealed a good agreement between simulations and observed data, as the individual variables, the dimensional reduction results, and the correlation structure between observed variables are properly captured.

## LTA model confirms known risk factors for severe post-COVID condition

As the model accurately describes the cohort-level dynamics, we next assessed the role of patient characteristics. While it is known that patient characteristics affect the chance of developing PCC after the acute infection[10], there is limited information on how they impact specific patient trajectories. LTA allows for a dependency of initial state and transition probabilities on patient characteristics. Specifically, we use a hierarchical model, which begins by using estimated values $\rho^{initial}$ and $\rho^{trans}$ to model the dependence on intermediate variables for each patient. These intermediate variables $r^{initial}$ and $r^{trans}$, then modulate the corresponding initial state and transition probabilities, which impact the patient trajectories. This hierarchical approach reduces the number of parameters compared to a model accounting for a direct dependence of initial state and transition probabilities of the patient characteristics. By inspecting how a given patient's characteristics change the probability of being in specific states, we uncover how varying these intermediate values ultimately drives patient state trajectories throughout the follow-up.

The meaning of directionality for the transition probability covariates is interpreted by inspecting the steady state distribution of the corresponding transition matrix constructed (Fig. 5b). Worsening, defined as a decreased probability of being in the Healthy state and an increased probability of being in the Respiratory or Fatigue PCC states, is correlated with female sex, age 41–60 and age > 60, corticosteroid therapy, and chronic respiratory disease (Fig. 5a). Infections at later waves are associated with higher probabilities to enter the Healthy state. The state distribution shown corresponds to the distribution of

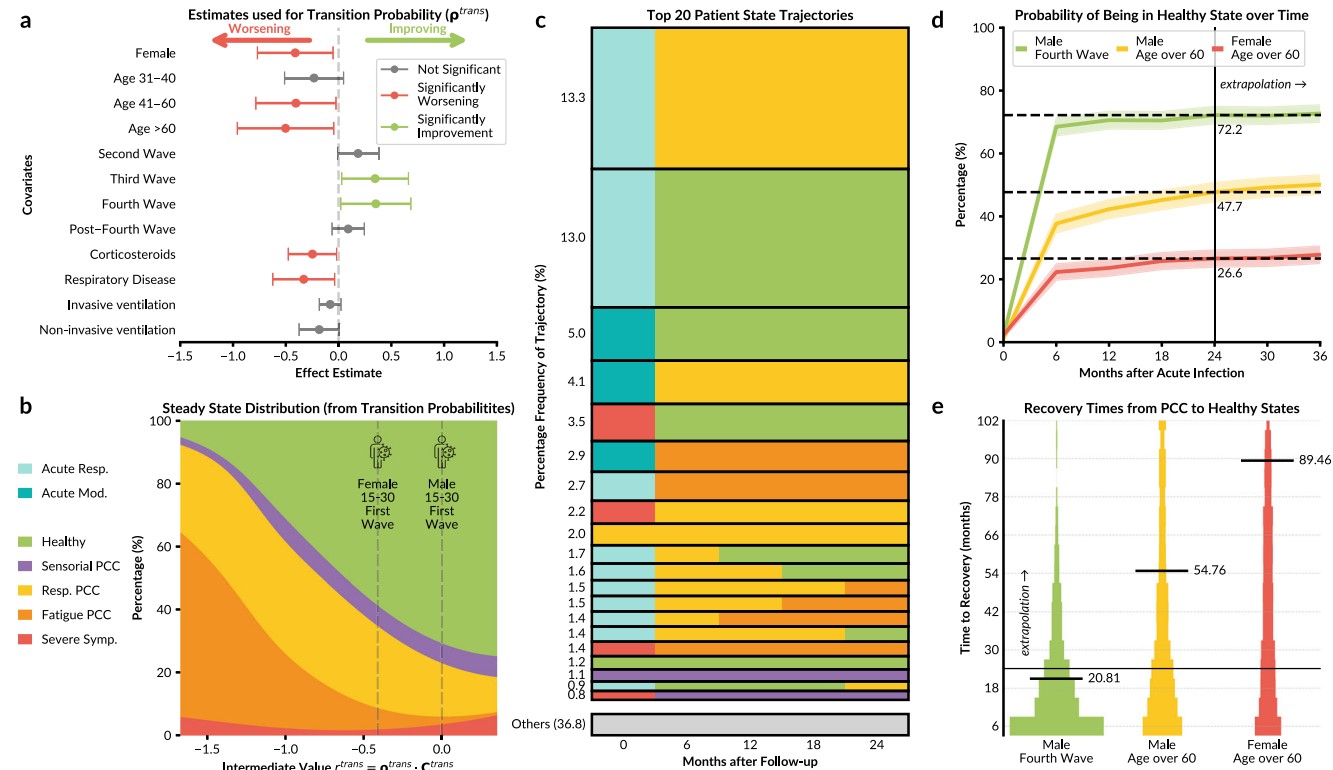

**Fig. 5 | Patient trajectories and their dependence on covariates. a** Forest plot for the dependence of transition probabilities on the covariates. Wald tests were used, where all p-values were two-sided. No adjustment for multiple comparisons was applied. Point estimates with 95% confidence bands (based on the parameter variance-covariance matrix). The bands show the propagated uncertainty of the estimated parameters. Coloured points are indicative of statistically significant (p-value ≤ 0.05) covariates leading to improving states if green (increased probability of Healthy state), while red indicates worsening states (increased probability of Respiratory/Fatigue PCC states). **b** Probability of state occurrence of patients with varying characteristics, with vertical lines representing an example male, aged 15 to 30, patient infected during the first wave, to indicate their respective probability distributions compared to a female with the same characteristics. **c** The 20 most frequently observed patient trajectories with corresponding percentage frequency after 1000 simulations using the patient of the real ORCHESTRA dataset. **d** Predicted probability of being in the Healthy State of various patients, with the solid line representing the mean of the simulations and the shaded band representing 95% coverage. **e** Predicted time for patients to transition from any of the PCC states to the Healthy state after 1000 simulations.

states in the distant future, which is shown to be similar in pattern to that at the 24-month follow-up (Supplementary Fig. 10).

The model shows that individual covariates do not meaningfully change the probabilities of being in a particular state at the acute stage of infection, as all covariate estimates have low statistical significance with large confidence bands. (Supplementary Fig. 11). However, the model is able to identify a distinct pattern of how covariates impact the stationary state distribution, despite this not being explicitly determined by the LTA model. There is a clear progression of improving stationary state distributions, as the Fatigue PCC state dominates initially for negative values of $r^{trans}$, with a higher probability to enter the Healthy state as the value increases (Fig. 5b).

Our model aligns with previous key findings on PCC[21]; that is, being female (0.41 ± 0.36), older age (0.40 ± 0.18 and 0.5 ± 0.45 for ages 41–60 and >60, respectively), and chronic respiratory disease (0.33 ± 0.29) lead to an increased risk of developing PCC. The third and fourth waves (−0.35 ± 0.32 and −0.35 ± 0.33, respectively) had significantly better recovery, compared to the first wave (Fig. 5a and Supplementary Table 5). The significance of corticosteroids used to treat COVID-19 at the acute infection (0.25 ± 0.23), indicating worsening health, adds to the growing complexity of the impact of the treatment for PCC. Indeed, as corticosteroid use was limited to patients with respiratory failure[24], the interpretability of the impact in the final model remains limited.

To understanding the actual phenotype trajectories of patients, rather than mean state pathways, we examined the frequency of observed state trajectories within the ORCHESTRA cohort

population (Fig. 5c). The assessment shows that the most common trajectories are dominated by starting in one of the Acute states and transitioning to either the Healthy or Respiratory PCC states, as most patients are not expected to develop severe PCC symptoms. Furthermore, after entering the Healthy, Respiratory PCC, Fatigue PCC, or Sensorial PCC states, no additional change of state is usually observed. Patients in the Acute Respiratory state followed by the Respiratory PCC state either enter the Healthy state or the Fatigue PCC state. The Sensorial PCC state was uncommon in the ORCHESTRA dataset but stable, as patients either began in the Sensorial PCC state at the acute stage or transitioned into it from the Severe Symptoms state. Notably, the Respiratory and Fatigue PCC states did not appear for this type of trajectory at any timepoint. Importantly, in the 20 most frequent trajectories, the Severe Symptom state occurs once during the acute infection and remains uncommon during the follow-up.

Simulations of the model also provide predictions for the impact of patient characteristics on recovery probability and the recovery time. Notably, the combination of male sex, ages 15–30 and infected at the fourth wave gives a high probability of being in the Healthy state at 24 months after acute infection (72.2%) while females aged over 60 had a remarkably lower probability (26.6%) (Fig. 5d). We also calculated the time to transition to the Healthy state, indicating recovery, for patients who begin at either Fatigue PCC, Respiratory PCC or Severe Symptom states (Fig. 5e). On average, females over 60 required more than double the time to recover (89.46 months), compared to males aged 15–30 infected at the fourth wave (20.81 months).

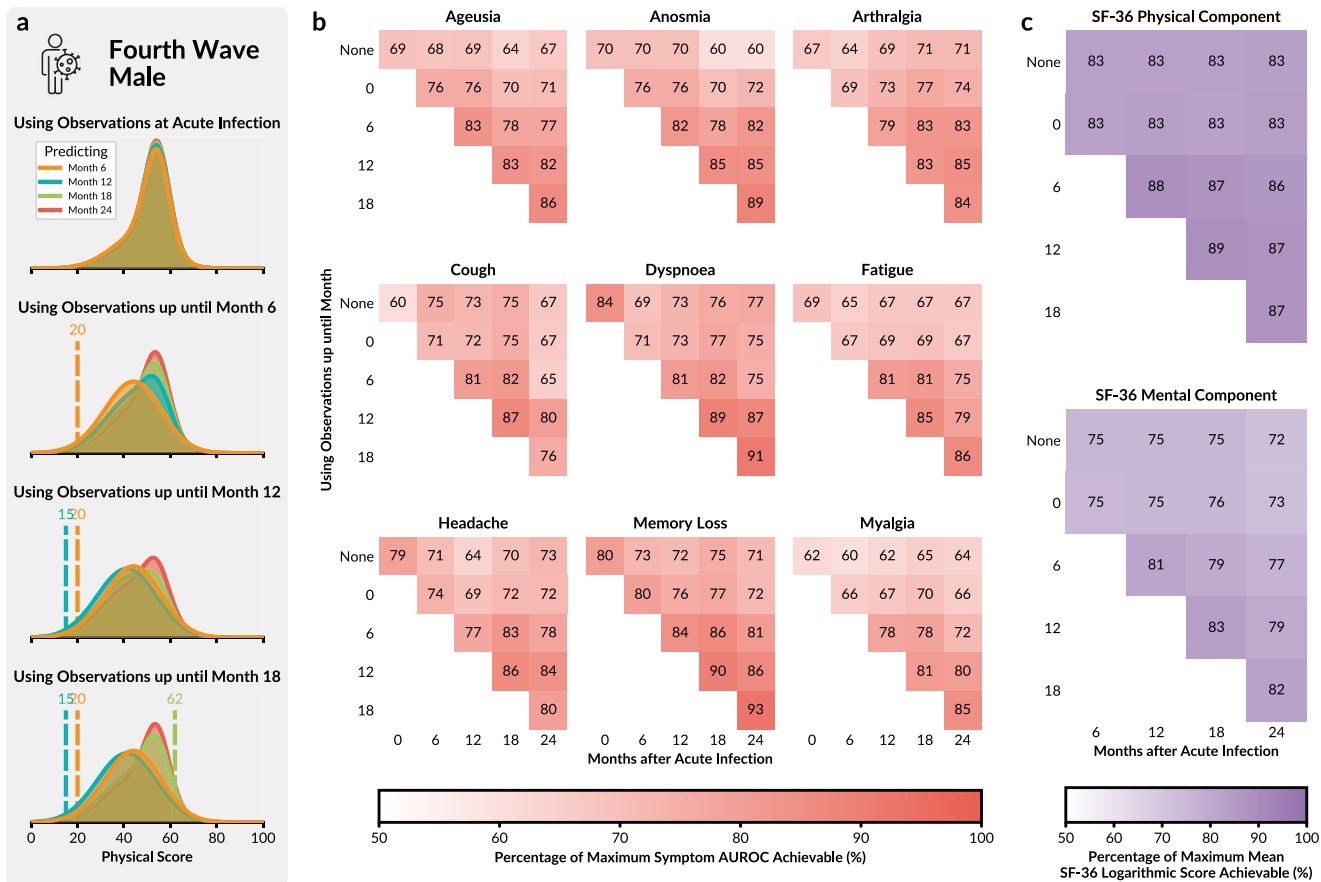

**Fig. 6 | Prediction accuracy of model for individual patients. a** Illustration of the prediction task for the physical score of a patient, using information up to different timepoints (individual rows). The predictions for different timepoints are indicated using different colours. As the patient's score (vertical dashed lines) is used in updating the current state probabilities, the probability distributions of the prediction change with the addition of new information. For this particular patient, the characteristics suggest a high probability of full recovery; yet, the recording of a poor SF-36 physical score at 6 and 12 months indicates that the patient is on a different trajectory, resulting in an update to the prediction for later timepoints. **b** The percentage of maximum achievable AUROC values of the 9 PCC-related symptoms at each timepoint after acute infection. Patient trajectories and predictions are updated by increasingly using previous observations up to the previous timepoint. **c** The average logarithm of the probability of mental/physical score given the estimated probability density for each patient across the timepoints and expressed as a percentage of the maximum achievable mean logarithmic score.

Overall, our in-depth analysis of the data-driven model reveals that it recapitulates known risk factors. Furthermore, the impact of risk factors on the initial and transition probabilities can be clearly interpreted. Most notably, it is found that patient Healthy and PCC states remain stable over time, rarely moving between follow-ups.

### Hidden Markov model enables longitudinal data integration for the improvement and personalisation of predictions

As the LTA provides an interpretable description of observations and risk factors at the cohort level, we next assessed its ability to provide personalised, patient-level predictions. Given the dynamic nature of the model, we consider not only the task of forecasting patient trajectories based on observations from the acute phase – a task previously recognized as challenging[21] – but also how additional observations along the patient trajectory can improve predictions for subsequent timepoints. This approach represents a filtering problem, leveraging the dynamic characteristics of the model and sequential patient data, without the need to re-estimate the entire model (see Utilising Patient History and Evaluation Metrics). We expect that the incorporation of symptom observations at intermediate timepoints increases confidence in identifying the correct state at those moments, thus enhancing the accuracy of state predictions for future visits (Fig. 6a).

The model demonstrates a good predictive performance of the ORCHESTRA patient cohort. In general, the best performance is achieved when the full information about the timepoint is utilised (Fig. 6b, c). There are only a few occurrences in which additional information decreases performance, such as the prediction of the presence of cough at month 24 using symptom and HRQoL measurements up until month 12 versus month 18 (80% to 76%). These exceptions are probably related to sampling noise and the stochasticity of the underlying process, as they occur only at Month 24 - the timepoint for which the fewest patients have been evaluated.

We assess the model's capability of predicting symptoms at each timepoint, updating the model to use all observations up to the evaluated follow-up. Across all timepoints, the LTA model predicts symptoms, updating the model to use all observations up to the evaluated follow-up, with an average Area under the Receiver Operator Curve (AUROC) score of 0.69 when evaluated on the entire ORCHESTRA dataset, while 5-fold cross-validation of the same model yielded a mean value of 0.65 (Supplementary Figs. 8 and 9).

In summary, our model not only provides strong agreement with the ORCHESTRA cohort dataset on an aggregated level, but also good agreement in predicting patient-specific symptoms and HRQoL scores in both in- and out- of sample datasets. The model accommodates for adding the most updated information using a filtering scheme rather than a complete re-estimation, which would be computationally burdensome.

## Discussion

Understanding the long-term consequences of infectious diseases requires comprehensive and flexible data analysis approaches, especially in situations where prior clinical knowledge is limited or evolving. This is particularly critical in the early stages of emerging diseases, such as during pandemics, when the underlying biological mechanisms, disease states, and symptom trajectories are not yet well understood. However, even in more established contexts, reliance on predefined disease states or symptom clusters can introduce bias and obscure subtle or atypical patterns present in complex data. There is a growing need for unbiased, data-driven methods that allow health states to be inferred directly from observed data, with methods that can robustly characterise disease heterogeneity and capture the evolving dynamics of PAIS such as Post-COVID Condition (PCC).

In this work, we demonstrate how Latent Transition Analysis – a powerful, well-established modelling framework based on Hidden Markov Models – can be extended and applied to long-term sequelae studies in a flexible and generalisable way. By enabling the (i) joint modelling of binary and continuous data, (ii) incorporation of trajectories with missing observations, and (iii) analysis of large numbers of covariates, LTA allows for the identification of latent health states and transition pathways, without requiring prior assumptions about disease structure or temporal dynamics.

We apply the LTA framework to the ORCHESTRA Long-Term Sequelae Cohort, one of the most comprehensive cohorts with up to 24 months of post-infection follow-up. The newly added 18- and 24-month assessments significantly expand the temporal resolution of the dataset, although, as expected, participation rates decline over time. While the decline in participation numbers is a limitation of the study, with only 419 individuals completing all five timepoints, the flexibility of our LTA framework allows us to leverage these uneven data patterns without the need for imputation. Hence, it provides a principled way to extract a signal despite missingness. Through systematic model selection and covariate-aware inference, we show that a fully connected 7-state HMM effectively captures the latent dynamics of PCC. The model recovers distinct phenotypic states, including a Healthy, Severe Symptom, and intermediate states such as Sensorial, Fatigue, and Respiratory PCC – each with specific symptom profiles and HRQoL patterns. The model also identifies two Acute-phase states, which are only present during the early stages of infection. Notably, the symptom clusters of the Acute Infection and PCC states, as well as the corresponding trajectories, emerge in a fully data-driven fashion, without any clinical labelling or hard-coded structure, demonstrating the plausibility of our modelling approach.

Our analysis revealed that LTA based on flexible HMMs allows for the data-driven definition of Acute Infection and PCC states. Model selection allows for robust assessment of the number of distinct states, and subsequent model simulations can be used to identify transition patterns between states.

Our analysis confirms known features of PCC while offering novel insights. LTA largely recreates the earlier PCA-derived phenotypes[21], and the overall picture emerging from the scientific literature[10,25]. In addition to the established research, the LTA revealed that fatigue underpins all PCC phenotypes. While our previous analysis identified the Sensorial PCC state with elevated rates of anosmia and ageusia at 12 months, the LTA model helps us to find that the state is also associated with reduced physical but preserved mental HRQoL, suggesting that sensory impairments may not significantly impact mental well-being. Conversely, the Fatigue PCC state involves broader symptom involvement and more marked reductions in both mental and physical HRQoL. We also observe that transitions between latent states are influenced by individual characteristics, such as age, sex, and the infection wave. While male patients from later waves tend to recover more rapidly, older individuals and females exhibit greater persistence in PCC states, especially the Fatigue and Respiratory PCCs. In comparison to earlier publications, which identified similar risk factors[10], the LTA describes in detail how these characteristics specifically impact patients' trajectories throughout the follow-up. Even if current definitions of PCC include fluctuation of symptoms[2], the main change in LTA states is the resolution of PCC.

Importantly, our model captures these individualised, covariate-dependent disease trajectories without introducing a prohibitive number of parameters. A key contribution of this work is the parsimonious parameterisation of covariate effects, which avoids the rapid growth in the number of parameters typically associated with covariate-dependent HMMs. Indeed, a classical model accounting for the dependence of all initial state and transition probabilities on all covariates would have required more than 1000 parameters to be estimated during model selection and could not have been handled using our available computational infrastructure. By projecting individual covariates into a low-dimensional scalar summary, we achieve interpretable, patient-specific transition probabilities while maintaining computational efficiency.

The model's predictive performance is strong, with good AUROC scores and reliable forecasts of both binary symptoms and continuous HRQoL metrics at future timepoints. While other clustering models of PCC measure performance on the ability to identify formed clusters[26], which in themselves are estimated via unsupervised algorithms, our model accounts for the uncertainty in the formation of clusters in the emergence of latent states in order to predict observable outcomes at each timepoint.

Despite its strengths, our study is subject to several limitations. Firstly, there are data collection constraints due to pandemic stress on health systems. Importantly, the majority of patients involved in our study were either admitted to general or ICU hospital wards at the acute stage of infection (75.4%), which is not representative of the general European population who were hospitalized from SARS-CoV-2[27]. This implies that when utilising the model in conjunction with patients and their characteristics of the ORCHESTRA cohort, there may be a smaller number of trajectories ending in the Healthy state than when applying to the general population. In addition, patient attrition at the 18- and 24-month follow-ups reduces the sample size for late-stage trajectory analysis, increasing uncertainty in those periods, highlighting the difficulties in maintaining an active follow-up in an observational study. We assume a missing completely-at-random (MCAR) pattern in the dropout process, so if healthier patients are disproportionately underrepresented in late follow-ups, this could bias our trajectory estimates; this assumption is given by the fact that the rate of healthy patients in the last follow-up is decreasing from the previous one. Secondly, the model is also subject to the typical non-identifiability issues of HMMs, such as label switching[28], which may overinflate uncertainty in covariate estimates. However, careful initialisation via multiple warm optimisation starts and consistent interpretation across a varying number of latent states helped to mitigate these effects. Lastly, it is important to note that HRQoL estimates per state and timepoint represent the HRQoL of patients within each state; however, they should not be interpreted as a consequence of the state itself, as both initial states and subsequent trajectories may be endogenous to HRQoL. The underlying clinical status of individuals before infection, which is typically well captured by HRQoL, likely influenced both the acute presentation and subsequent PCC trajectories. As pre-infection HRQoL data are not available, this potential source of confounding cannot be fully accounted for.

Our open-source Julia implementation provides researchers with tools to fit, interpret, and visualize latent transition models with customizable covariates and state structures. It includes simulation tools, diagnostic routines, and an interactive visualization interface, enabling researchers to explore how changing covariate inputs alters individual trajectories. This makes our framework a ready-to-use solution for

other PAIS studies, particularly those involving incomplete, multimodal clinical data.

In short, the LTA paradigm manages to combine the benefits of both unsupervised clustering methods and clinically guided grouping of patient observations used for single timepoints, on longitudinal datasets. Unlike classical techniques, LTA facilitates a more holistic approach to PAIS phenotype identification, as the relationship between observations and hidden states is defined in an intuitive manner, allowing better accessibility and interpretability for trained clinicians while not sacrificing the necessary complexity of the model. Despite the inherent limitations of our PCC study, LTA enabled an analysis of patient-level trajectories over time. The framework supports patient-level prediction of future state trajectories and recovery times at any point after the acute infection phase, by leveraging the distribution of symptoms observed at previous timepoints. LTA constitutes a data-driven method that allows health states to be inferred directly from observed data while relying on a few underlying model assumptions that can robustly characterise disease heterogeneity and capture the evolving dynamics of long-term conditions. Thus, this generalised approach is not specific to disease progressions of PCC in the ORCHESTRA cohort study only, but offers practical utility for clinical monitoring and therapeutic development of all longitudinal PAIS studies characterised by a broad spectrum of symptoms that persist or emerge after the initial resolution of infection. These syndromes present substantial challenges to both research and clinical care due to their protracted nature, heterogeneity, and often unclear pathophysiology. The framework is easily extendable to general PAIS studies that include relevant biomarkers, patient-reported outcomes, and structured assessments such as clinical scales or omics profiles, which enables the identification of multimodal latent phenotypes that reflect disease biology across diverse data types. Future work may also include time-varying covariates or explicit modelling of interventions (e.g., post-infection vaccination or rehabilitation), enabling causal inference about recovery-modifying treatments.

In summary, we present an extended LTA framework capable of uncovering the hidden dynamics of the progression of post-acute infection syndromes in a data-driven manner. Applied to the ORCHESTRA cohort, the model identifies meaningful PCC phenotypes, explains patient heterogeneity in recovery, and provides predictive insight into individual trajectories – despite challenges such as missing data and model ambiguity. We identified seven states describing Acute Infection and PCC, and the most common trajectories, being able to predict PCC resolution based on patient-level covariates. By eliminating the need for prior disease knowledge or symptom grouping assumptions, our approach is particularly valuable in emerging or poorly understood conditions, offering a robust and generalizable tool for post-infection surveillance, cohort studies, and future pandemic preparedness.

## Methods
### Study design and participants
The ethics committees of the University of Verona (UNIVR), University of Bologna (UNIBO), Institut National de la Santé et de la Recherche Médicale (INSERM), Andalusian Health Service (SAS), and University Medical Center Groningen (UMCG) all approved the study protocol. The study was conducted in accordance with the principles of the Declaration of Helsinki. Each patient enroled signed a written-informed consent form.

The ORCHESTRA Long-term Sequelae Cohort (CT registration number: NCT05097677) comprises five prospective subcohorts from 56 centres across four countries (France, Italy, the Netherlands, and Spain). Eligible participants included both hospitalised and non-hospitalised patients aged over 14 years, with laboratory-confirmed SARS-CoV-2 infection, enroled after providing written

informed consent. Participants were systematically followed at 6-, 12-, 18-, and 24-months after infection in outpatients' clinics or at the patients' home for all the centres. Each follow-up involved clinical assessments by qualified medical personnel and extensive laboratory testing; additional testing was performed if clinically needed. Nasopharyngeal swabs were collected at baseline for diagnosis and to identify the variant of concern (VoC) and were repeated only if positive after 30 days from the initial diagnosis. VoC typing and serological analyses were performed at the Antwerp laboratory or in local laboratories following standardised protocols. Local cohorts datasets that started before December 2020 were homogenised and standardised as previously described[29,30].

Data collected at baseline included date of symptom onset and diagnosis, duration of symptoms, demographic characteristics, comorbidities, clinical presentation, hospitalisation, admission to ICU, treatment, and post-acute infection complications. Recommendations for early antiviral treatment (e.g., anti-SARS-CoV-2 therapy within the first five days of onset of symptoms, according to national recommendations) included three anti-SARS-CoV-2 spike monoclonal antibodies (bamlanivimab, bamlanivimab/etesevimab, and casirivimab/imdevimab). HRQoL was assessed through the physical component score and the mental component score of the SF-36 questionnaire[31,32] at 6-, 12-, 18-, and 24-months after infection.

Study data were collected and managed using the REDCap electronic data capture tool (Research Electronic Data CAPture)[33]. Since the cohorts in France and the Netherlands started before the ORCHESTRA project was financed (in February and March 2020, respectively), data from these two cohorts went through a post-data collection harmonisation process under the supervision of the Charité, Universitäts Medizin Berlin, and the Centre Informatique National de l'Enseignement Superieur[29,34].

The SF-36 questionnaires were scored using the PRO CoRE software developed by QualityMetrics, which applies US1998 norms. The threshold for suboptimal scores (50) represents the norm-based scoring mean of the 1998 US general population (mean=50, SD=10). Scores below this threshold indicate below-average HRQoL compared to the reference population.

### Hidden Markov Model
We assume that a patients' collection of observations (occurrence of symptoms and HRQoL scores) is driven by a patient's respective unobserved PCC phenotype/severity, which may change over time as patients move towards more severe or healthy states. We model this process by a discrete-time and discrete-space hidden Markov model (HMM) with non-zero $N \in \mathbb{Z}^+$ latent states. Let $S_i$ be the $i$-th latent state, where $i \in \{1, 2, ..., N\}$ with $T$ as the timepoint of the last of observation. The probability of the state occurring at a given discrete timepoint, $t \in \{1, 2, ..., T\}$, is $P(S_i^t = 1)$. Importantly, a patient may only occupy one state at any time.

Considering one patient who has $l \in \{1, 2, ..., L\}$ individual observations (such as anosmia or mental HRQoL score), so that $\mathbf{X}^t \in \mathbb{R}^L$ is the observed vector at time, $t$, where $\mathbf{X}^t = (X_1^t, X_2^t, ..., X_L^t)$ explicitly. For the same given patient, the patient characteristics are encapsulated in two covariate vectors, $\mathbf{C}^{initial} \in \mathbb{R}^{K_{initial}}$ which are used to determine the initial state probability distribution for the given patient's covariates, and the other, $\mathbf{C}^{trans} \in \mathbb{R}^{K_{trans}}$ which impact a patient's corresponding transition state probabilities.

The HMM allows us to model the progression of PCC as the movement of PCC severity state from one timepoint to the next in 6-month intervals. The $i$–th state is responsible for a particular probability, $b_{il}$ of manifesting the $l$-th discrete observed symptom, which are collectively defined as the emission probabilities of the binary symptom observations. The continuous HRQoL scores are modelled as coming from a Gaussian distribution with mean, $\mu_{il}$ and covariance, $\sigma_{il}$

associated with the $i-$th state,

$$X_l = 1 \mid S_i = 1 \sim Bernoulli(b_{il}) \ if \ observation \ X_l \ is \ binary, \qquad (1)$$

$$X_l = x \mid S_i = 1 \sim Normal(\mu_{il}, \sigma_{il}) \ if \ observation \ X_l \ is \ continuous. \qquad (2)$$

When considering the PCC study presented in this work, for each given state, $S_i$, there is an associated probability of a patient manifesting one of the $L = 11$ observations, composed of 9 binary symptoms and 2 HRQoL scores (physical and mental). We note that the parameters defining the emissions ($b_{il}$, $\mu_{il}$ and $\sigma_{il}$) are independent of time, whereas the binary symptom observations and continuous HRQoL scores are dependent. This is because observations themselves only depend on the current state, where the state probability changes from one time-point to the next. The probability of an observation $X_l^t$ for an individual in the $i$-th state is defined as

$$P(X_l^t \mid S_i^t = 1) \equiv c_i(X_l).$$

Assuming conditional independence among observations at a specific time point, the joint probability of a set of observations at this time point is obtained as the product of their individual probabilities, conditioned on the occurrence of the $i-$th state.

As the severity state of a patient can change within the follow-up time period, there is an associated transition rate, $a_{ij}$, which is the probability of moving from the $i$-th state to the $j$-th state, dependent on a patient's associated covariates, $\mathbf{C}^{trans}$. The matrix $a_{ij}$ is row-stochastic, and so the $i-$th row has $N-1$ degrees of freedom as it must sum to 1, and there are total $N$ rows. To ensure that these transition probabilities are bounded between 0 and 1 while being dependent on a patient's covariates, we parametrise each row of $a_{ij}$ by a multinomial logistic regression function. Thus, the logistic function parametrising the transition probabilities has regression coefficients $\beta_{hij}^{trans} \in \mathbb{R}^{2 \times N \times (N-1)}$, where the function consists of 2 components; the intercept $\beta_{1ij}$ and the coefficient, $\beta_{2ij}$ used in conjunction with the covariate term. In order to capture the unique hidden state trajectory of each patient as well as ensure interpretability of covariate effects on the transition rates, we introduce the vector, $\boldsymbol{\rho}^{trans}$, which reduces the effect of the vector of covariates, to a single value, $r^{trans}$, through the inner product $r^{trans} = \boldsymbol{\rho}^{trans} \cdot \mathbf{C}^{trans}$. This scalar value is then used as the single independent variable in the regression function. Thus, the transition probabilities are defined as,

$$P(S_j^t = 1 \mid S_i^{t-1} = 1, \mathbf{C}^{trans}) \equiv a_{ij} = \begin{cases} \dfrac{1}{1 + \sum_{j=1}^{N} \exp(\beta_{1ij}^{trans} + \beta_{2ij}^{trans} r^{trans})} & if \ i = j, \\[4mm] \dfrac{\exp(\beta_{1ij}^{trans} + \beta_{2ij} r^{trans})}{1 + \sum_{j=1}^{N} \exp(\beta_{1ij}^{trans} + \beta_{2ij}^{trans} r^{trans})} & if \ i \neq j, \end{cases} \qquad (3)$$

where the reference state is the $i$-th state. We note that for a standard multinomial function, the number of estimated parameters associated for the transition probabilities, would be $N(N-1)K_{trans}$. Our hierarchical modelling approach reduces the number of parameters to $2N(N-1) + K_{trans}$ when considering sufficiently large models ($N, K_{trans} \geq 3$). This parameterisation reduces the complexity of possible trajectories of the patient as there is no longer a unique set of regression coefficients for each covariate for each $N-1$ states. However, the scalar value $r^{trans}$ now gives a more interpretable impact of the transition probabilities, $a_{ij}$, as the variation of a single value affects the entire transition matrix for a given patient and their covariates. A similar parametrisation is used for the initial state distribution with $r^{initial} = \boldsymbol{\rho}^{initial} \mathbf{C}^{initial}$, which calculates the probability of being in the $i$-th state at the first timepoint, $\pi_i = P(S_i^{t=0})$, using $i = 1$ as the reference state (Supplementary 1.1.1). Thus, a patient's covariates affect the probability of state occurrences at each timepoint, which in turn determines how likely the patient exhibits a given symptom or HRQoL score.

For ease of notation, we write the complete set of parameters, which define the reduced covariate-driven hidden Markov model, as $\theta = \{\boldsymbol{\rho}^{initial}, \beta^{initial}, \boldsymbol{\rho}^{trans}, \beta^{trans}, b_{il}, \mu_{il}, \sigma_{il}\}$ where we ensure $b_{il}, \mu_{il} \in [0, 1]$ and $\sigma_{il} \geq 0$ (Supplementary 1.1.2).

## Model Fitting and Selection

To fit a discrete-time, discrete-space HMM, the Baum-Welch algorithm is often utilised, being a special case of an expectation-maximisation (EM) algorithm[35,36] in order to estimate the model parameters that best describe the observational sequences of all patients. There have been extensions made[18] to define explicit update steps for each parameter when the initial and transition state probabilities are constructed from a collection of multinomial functions. Rather than deriving the explicit update rule of an EM algorithm for our covariate-dependent model, we construct the likelihood function efficiently using the forward algorithm of the Forward-Backwards algorithm[37], and utilising fast automatic differentiation tools to find the best model parameters in a gradient-based optimiser.

Using the iterative rule from the forward algorithm, the likelihood function is as follows:

$$\alpha_t(j) = \sum_{i=1}^{N} \alpha_{t-1}(i) a_{ij} c_j(\mathbf{X}^t), \qquad (4)$$

The iterative rule of the forward algorithm builds up the probability of the observed sequence step by step. At each timepoint $t$, the forward probability, $\alpha_t(j)$ represents the joint probability of being in the $j$-th state and having observed all data up to $t$. This is obtained by summing over all possible predecessor states, weighting their forward probabilities by the transition probability $a_{ij}$ and multiplying by the probability of observing a new collection of symptoms and HRQoL scores, $c_j(\mathbf{X}^t)$ (Supplementary 1.1.2).

By carrying this recursion forward, the algorithm accumulates the probability of all possible state paths consistent with the observations. At the final timepoint $T$, every possible path through the model has been incorporated, so summing $\alpha_T(j)$, across all states yields the total probability of the observed data under the model, which is equivalent to the likelihood. Thus, the likelihood of the set of parameters, $\theta$ of the HMM, given the observational sequence for all times, $\mathbf{X}^{1:T}$ for a single patient is,

$$L(\theta \mid \mathbf{X}^{1:T}) = \sum_{j=1}^{N} \alpha_T(j). \qquad (5)$$

We introduce two penalty terms that assist in finding strong starting parameters necessary to the optimal model. First, optimisation can be prone to degenerate solutions where infinite variance is estimated for the Gaussian distributions associated with each states, $\sigma_{il}$, leading to numerical instabilities, which are typically seen in Gaussian mixture models[38]. One proposed solution is to add another regularisation term dependent on the variance and scaled by the number of observations[39], thereby punishing large variances while preventing attraction towards zero as well. Second, to ensure $\boldsymbol{\rho}^{initial}$ and $\boldsymbol{\rho}^{trans}$ are consistently scaled with a magnitude equal to 1, we introduce $L2$ regularisation on the norm, $\|\boldsymbol{\rho} - 1\|^2$ for both vectors as a soft constraint for consistent interpretability across different models. Thus, the regularised log likelihood is as follows,

$$\mathcal{L}_{reg}(\theta \mid \mathcal{D}) = \sum_{d=1}^{D} \log \left( \sum_{j=1}^{N} \alpha_T^d(j) \right)$$
$$+ \lambda_1 \left( \|\boldsymbol{\rho}^{initial} - 1\|^2 + \|\boldsymbol{\rho}^{trans} - 1\|^2 \right) \qquad (6)$$
$$+ \lambda_2 \sum_{l \in \mathcal{C}} \left( \frac{1}{n} \sum_{i=1}^{n} \sigma_{il}^{-2} + \sum_{i=1}^{n} \log \sigma_{il}^2 \right),$$

where the first term is the log-likelihood of the dataset, $\mathcal{D}$ consisting of $d = \{1, 2, ..., D\}$ patients where now $\alpha_T^d(j)$ the forward path probability of the $d$-th patient. We note that $\mathcal{C} \subset 1, ..., L$ is the set of indices corresponding to continuous observations.

Although LTA models are typically applied to complete datasets where imputation methods may be employed, our model is fitted using a Full-Information Maximum Likelihood (FIML) approach in order to handle missing observations (either partially or completely missing) at any timepoint, assuming the observations are missing completely at random (MCAR). This is done by marginalising out the contribution to the likelihood when an observation is not known, which is equivalent to skipping the contribution entirely, as no new information is added to the forward path probability (Supplementary 1.1.4). Our FIML approach allows efficient usage of all information available within the ORCHESTRA Long Term Sequelae dataset without subjecting the model parameter estimates to possible biases introduced by selected imputation methods.

To estimate the optimal parameters, we employ the Broyden-Fletcher-Goldfarb-Shannon (BFGS) algorithm for optimisation[40], with gradient calculations completed using automatic differentiation[41]. A single optimisation run is completed in a 2-stage approach, where the parameters are first estimated with the regularisation terms set to $\lambda_1 = 10$ and $\lambda_2 = 1$. After convergence, the estimated parameter vector is used as the starting parameter vector for fitting, this time without the penalty terms. This process leads to a reduced number of degenerate solutions found, as well as easier comparisons of $\boldsymbol{\rho}^{initial}$ and $\boldsymbol{\rho}^{trans}$ across runs, as the magnitudes are roughly consistent. Latin Hypercube sampling was used to generate initial starting points for the majority of parameters[42], except for the mean and standard deviation of the Gaussian distributions, which were chosen from the clusters found in a $N$-means clustering step of the continuous variables, which ensure adequate starting values. Uncertainty of the estimated parameters is obtained via the associated variance-covariance matrix which is calculated by taking the inverse of the Fisher Information matrix. This is equivalent to taking the second derivative of the likelihood function evaluated at the parameter estimates[43].

Model selection was completed by first estimating the null HMM model for a given number of latent states, without the influence of covariates in the initial and transition state probabilities, using the 2-stage optimisation approach with 20 multistarts each. 20 starts were chosen due to available computation resources, but were seen to be sufficient in finding similar solutions across starts. Age and sex covariates are then added to describe the initial and transition probabilities, and the parameters are re-estimated with a warm start using the estimated parameters for the best-fitting null model for each number of states, $N$. A forward selection approach is then made, where a single covariate is introduced to the age/sex model, and a similar re-estimation is completed using a warm start. The corresponding Bayesian Information Criterion (BIC) value[44] is calculated after estimating the new covariate model, and the covariate is selected and added to the base model if it returns the lowest BIC value amongst all covariates that are currently not included in the model at the selection round. This process is repeated until an improvement of less than 10 occurs across all covariates. This was completed for 5 models (4 to 8 latent states), and we chose the final model using the BIC, as there is strong support for its usage in identifying the most suitable number of latent states[45-47].

In comparing models varying in size from $N = 4$ to 8 states using the forward selection, it was found that the $N = 7$ model gave the lowest BIC value in comparison to both the null (no covariates used) and forward-selected model groups (Supplementary Fig. 2). For the 7 state model, the following covariates were selected in the order mentioned: oxygen therapy, infection wave, chronic respiratory disease, and corticosteroids (Supplementary Fig. 4). These covariates were added consecutively to the base model that included age and sex already,

which resulted in the final 7 state model presented. A patient's history of smoking, as well as therapy history at the acute stage (usage of antivirals (remdesivir), monoclonal antibodies, immunomodulators, and vaccination before the acute infection) were also considered in the forward selection approach but were ultimately not selected. Hospitalisation was excluded from the list of possible covariates as it is strongly correlated with unrecorded medical complications, making the clinical meaning difficult to interpret. To evaluate the model performance of the final 7-state model, the chosen set of covariates was used in a 5-fold cross-validation procedure with 20 multistarts, each with a cold starting parameter vector and not utilising the 2-stage approach.

We used a lab-internal HPC cluster for model fitting and model selection. The cluster comprises eight CPU nodes with dual AMD EPYC 7443 (2.85 GHz) processors and six nodes with dual AMD EPYC 7F72 (3.20 GHz) processors. Five independent jobs were submitted for models with $N = 4$ to $N = 8$ latent states. Each job was allocated 24 cores and a wall time limit of 7 days. The total CPU time consumed across all jobs was approximately 1400 core-hours.

## Forward simulations

Using the fitted 7-state model, we performed 1000 simulations based on the dataset for the ORCHESTRA Cohort, by utilising each patient's covariates and predicting their corresponding symptom reports and HRQoL scores at each timepoint. We then compared these forward simulated trajectories against the observed data at each follow-up month to evaluate the model's performance in capturing both aggregate population-level statistics and predicted state probability distributions.

## Utilising patient history and evaluation metrics

The latent transition model with the set of estimated parameters, $\widehat{\theta}$ can predict the probability of the $l$th observation occurring at a given timepoint, $t$, $P(X_l^t | \widehat{\theta})$, which is computed by finding the patient's corresponding state distribution at $t$, utilising the initial state distribution and transition probabilities. However, if we would like to use previous observations in order to inform the prediction, then

$$P(X_l^t | \mathbf{X}^{1:t-1}, \widehat{\theta}) = \sum_{j=1}^{N} c_j(X_l^t) \sum_{i=1}^{N} a_{ij} P(S_i^{t-1} | \mathbf{X}^{1:t-1}, \widehat{\theta}). \quad (7)$$

So, we may improve our prediction of $P(X_l^t | \widehat{\theta})$ by simply updating a patient's state distribution, $P(S_i^{t-1} | \mathbf{X}^{1:t-1}, \widehat{\theta})$, which utilises the observations at 1 to $t - 1$ timepoints (Supplementary 1.1.3). If we expect our observations to be informative of future timepoints, then the predictive capability should likewise improve, with this having the benefit of utilising newly added patient history without complete re-estimation of the model parameters.

In order to evaluate how good these predictions of the observations are using the model and corresponding updates, the AUROC value is calculated for the binary observations. For the set of 9 PCC-related symptoms, at each timepoint we calculate an associated AUROC value using patient observations up until and including the timepoint before evaluation.

In order to evaluate how well the density predicted for the continuous HRQoL scores are, we calculate the mean logarithmic score for the mental and physical questionnaire scores at each timepoint, $\frac{1}{D}\sum_{d=1}^{D} \ln P(X_l^t = x^d)$, where $x^d$ is the observed value of the $d$th patient.

Similarly, we may utilise the distribution of states at time, $t$, informed by observations up to and including $t$ now, to predict observations at the very same timepoint,

$$P(X_l^t | \mathbf{X}^{1:t}, \widehat{\theta}) = \sum_{j=1}^{N} c_j(X_l^t) P(S_j^t | \mathbf{X}^{1:t}, \widehat{\theta}). \quad (8)$$

Thus, we can obtain a maximum AUROC and log mean score value achievable of the estimated model, as the observations at $t$ provide the best determination of the states at $t$, if we do not consider future observations informing the probability of the states. Thus, we express the performance of our predictions as a percentage of the maximum achievable AUROC for binary symptoms and the maximum achievable mean log score for the continuous observations.

## Reporting summary

Further information on research design is available in the Nature Portfolio Reporting Summary linked to this article.

## Data availability

The complete ORCHESTRA WP2 Long Term Sequalea dataset is not openly available due to the sensitivity of the data, but can be requested via an online form specifying the cohort data of interest, the variables, and the research question to pursue at https://dataportal.orchestra-cohort.eu/data_access/. It is possible to directly access a subset of anonymized data via the public use file at https://dataportal.orchestra-cohort.eu/public_use_file.

## Code availability

R 4.3.3 was primarily used for the preprocessing of the dataset. All data formatting for input into the model, simulation, model implementation, model selection, performance evaluation and exportation of results were conducted in Julia version 1.10.1[48], primarily using the Optim package version 1.13.2 for the optimisation procedure[49]. Plotted figures were created in Python 3.10.12 using Matplotlib 3.6.3. The implementation can be found online on https://doi.org/10.5281/zenodo.17787061.

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

## Acknowledgements

The ORCHESTRA project has received funding from the European Union's Horizon Europe Research and Innovation Programme under Grant Agreement No. 101016167. Views and opinions expressed are however, those of the author(s) only and do not necessarily reflect those of the European Union or of the European Research Executive Agency (REA). Neither the European Union nor the granting authority can be held responsible for them. This work was also supported by the Deutsche Forschungsgemeinschaft (DFG, German Research Foundation) under Germany's Excellence Strategy (EXC 2047–390685813, EXC 2151–390873048), and by the University of Bonn (via the Schlegel Professorship) [Jan Hasenauer].

## Author contributions

Lorenzo Maria Canziani (L.M.C.), Elisa Gentilotti (E.G.), Anna Górska (A.G.), Roy Gusinow (R.G.), Jan Hasenauer (J.H.) and Evelina Tacconelli (E.T.) were involved in the conceptualisation of the project. Carolina Alvarez Garavito (C.A.G.), A.G., R.G., J.H., Iris Lopes-Rafegas (I.L.), Elisa Sicuri (E.S.) and E.T. were responsible for the methodology. A.G., R.G., and I.L. handled the programming of the software. Maria Giulia Caponcello (M.G.C.), Michela Di Chiara (M.D.C.), Aline-Marie Florence (A.F.), E.G., Jade Ghosn (J.G.), Maddalena Giannella (M.G.), Cédric Laouénan (C.L.), Nadhem Lahfej (N.L.), Fulvia Mazzaferri (F.M.), Zaira R. Palacios-Baena (Z.R.P.), Lidia Del Piccolo (L.D.P.), Adriana Tami (A. Tami), Alice Toschi (A. Toschi), Karin I. Wold (K.I.W.) and the ORCHESTRA Study Group were involved in patient recruitment and data collection. L.M.C., M.G.C., M.D.C., E.G., A.G., R G., N.L., Elisa Rossi (E.R.) and A. Tami managed and curated the data. L.M.C., A.G., R.G., J.H., I.L. and E.T. were involved in the initial writing of the original draft, while all authors were involved in the reviewing and editing of the final version. L.M.C., A.G., R.G., J.H. and I.L. aided in the conceptualisation and visualisation of the figures.

## Funding

## Competing interests

The authors declare no competing interests.

## Additional information

[1]The Life and Medical Sciences Institute (LIMES), University of Bonn, Bonn, Germany. [2]Bonn Center for Mathematical Life Sciences, University of Bonn, Bonn, Germany. [3]Division of Infectious Diseases, Department of Diagnostics and Public Health, University of Verona, Verona, Italy. [4]ISGlobal, Barcelona, Spain. [5]Facultat de Medicina i Ciències de la Salut, Universitat de Barcelona (UB), Barcelona, Spain. [6]University of Groningen, University Medical Center Groningen,

Department of Medical Microbiology and Infection Prevention, Groningen, The Netherlands. [7]LSE Health, London School of Economics and Political Science, London, UK. [8]Centro de Investigação em Saúde de Manhiça (CISM), Manhiça, Mozambique. [9]APHP Nord, Hôpital Bichat, Service des Maladies Infectieuses, Paris F75018, France. [10]Université Paris Cité, INSERM UMR 1137 IAME, Paris F75018, France. [11]APHP Nord, Hôpital Bichat, Department of Epidemiology Biostatistics and Clinical Research, Paris, France. [12]Department of Neurosciences, Biomedicine and Movement Sciences, University of Verona, Verona, Italy. [13]Department of Medical and Surgical Sciences, Alma Mater Studiorum, University of Bologna, Bologna, Italy. [14]Infectious Diseases Unit, Department for Integrated Infectious Risk Management, IRCCS Azienda Ospedaliero-Universitaria di Bologna, Bologna, Italy. [15]Unidad Clnica de Enfermedades Infecciosas y Microbiologa, Hospital Universitario Virgen Macarena; Departamento de Medicina, Universidad de Sevilla, Instituto de Biomedicina de Sevilla (IBiS)/CSIC, Seville, Spain. [16]CIBERINFEC, Instituto de Salud Carlos III, Madrid, Spain. [17]CINECA Interuniversity Consortium, Bologna, Italy. [32]These authors jointly supervised this work: Evelina Tacconelli, Jan Hasenauer. ✉e-mail: evelina.tacconelli@univr.it; jan.hasenauer@uni-bonn.de

## the ORCHESTRA study group

**University of Verona (UNIVR)** Elena Addis[3], Maddalena Armellini[3], Anna Maria Azzini[3], Benedetta Barana[3], Lucia Bonato[3], Elena Carrara[3], Alessandro Castelli[3], Filippo Cioli Puviani[3], Michela Conti[3], Raffaella Cordioli[3], Carmine Cutone[3], Ruth Joanna Davis[3], Pasquale De Nardo[3], Miriam Emiliani[3], Alessio Esposito[3], Daniele Fasan[3], Giada Fasani[3], Giorgia Franchina[3], Jacopo Garlasco[3], Enrico Gibbin[3], Salvatore Hermes Dall'O'[3], Chiara Konishi De Toffoli[3], Lorenza Lambertenghi[3], Federico Lattanzi[3], Andrea Leonardi[3], Francesco Luca[3], Gaia Maccarrone[3], Massimo Mirandola[3], Matteo Morra[3], Alessandra Nazeri[3], Matilde Rocchi[3], Giulia Rosini[3], Chiara Perlini[3], Maria Diletta Pezzani[3], Laura Rovigo[3], Anna Giulia Salvadori[3], Andrea Sartori[3], Alessia Savoldi[3], Rebecca Scardellato[3], Marcella Sibani[3], Erica Sodano[3], Simona Sorbello[3], Lorenzo Tavernaro[3], Giorgia Tomassini[3], Alessandro Visentin[3], Stefania Vitali[3], Andrea Volpe[3], Chiara Zanchi[3], Gloria Mazzali[18], Giovanni Stabile[18], Gianluca Vantini[18], Riccardo Cecchetto[19], Davide Gibellini[19], Nicolò Cardobi[20], Debora Calì[12], Maria Paola Cecchini[12], Maddalena Marcanti[12], Anna Mason[12], Salvatore Monaco[12], Marco Pattaro Zonta[12], Cinzia Perlini[12], Gianluigi Zanusso[12], Elda Righi[21], Mariana Nunes Pinho Guedes[15,22], Maria Mongardi[23,24], Concetta Sciammarella[25], Claudio Micheletto[26] & Paolo Gisondi[27]

**University of Bologna (UNIBO)** Natascia Caroccia[14], Cecilia Bonazzetti[13,14], Beatrice Tazza[14], Zeno Igor Adrien Pasquini[14], Domenico Marzolla[13], Giacomo Fornaro[14], Fabio Trapani[14,28], Lorenzo Marconi[14], Luciano Attard[14], Sara Tedeschi[13,14], Silvia Vituliano[13], Liliana Gabrielli[13,29] & Tiziana Lazzarotto[13,29]

**Servicio Andaluz de Salud (SAS)** Jesús Rodrguez-Baño[15,16], Mara Isabel Garcia Sánchez[30], Ana Belén Hidalgo Céspedes[30], Aurora Aleman Rodriguez[15,16], Lola Cubero Aranda[15,16], Paula Olivares Navarro[15,16], Sandra De la Rosa Riestra[15,16] & José M. Bravo-Ferrero[15,16]

**University Medical Center Groningen (UMCG)** Gerolf de Boer[6], Bernardina T. F. van der Gun[6], Mara F. Vincenti-González[6], Alida C. M. Veloo[6], Daniele Pantano[6], Margriet van der Meer[6], Lilli Gard[6], Erley F. Lizarazo[6], Marjolein Knoester[6], Alex W. Friedrich[6] & Hubert G. M. Niesters[6]

**CINECA** Salvatore Cataudella[31] & Chiara Dellacasa[31]

**University of Bonn** Manuel Huth[1,2] & Clemens Peiter[1,2]

[18]Division of Geriatrics, Department of Medicine, University of Verona, Verona, Italy. [19]Division of Microbiology, Department of Diagnostics and Public Health, University of Verona, Verona, Italy. [20]Department of Diagnostics and Public Health, University of Verona, Verona, Italy. [21]Guys and St Thomas's NHS Foundation Trust, London, UK. [22]Infection and Antimicrobial Resistance Control and Prevention Unit, Hospital Epidemiology Centre, Centro Hospitalar Universitário São João, Porto, Portugal. [23]ANIPIO, Società Scientifica Nazionale degli Infermieri Specialisti del Rischio Infettivo - National Association of Nurses for the Prevention of Hospital Infections, Bologna, Italy. [24]Department of Medicine, University of Parma, Parma, Italy. [25]Division of Pathology, Department of Diagnostics and Public Health, University of Verona, Verona, Italy. [26]Pulmonary Unit, Integrated University Hospital of Verona, 37129 Verona, Italy. [27]Section of Dermatology and Venereology, Department of Medicine, University of Verona, Verona, Italy. [28]Infectious Disease Unit, Department of Oncology and Hematology, Guglielmo da Saliceto Hospital, Piacenza, Italy. [29]Microbiology Unit, IRCCS Azienda Ospedaliero-Universitaria di Bologna, Bologna, Italy. [30]Biobanco del Sistema Sanitario Público de Andaluca, Nodo del Hospital Universitario Virgen Macarena, Seville, Spain. [31]HPC Department, CINECA Consorzio Interuniversitario, Bologna, Italy.

