## [Transparent Peer Review file · Nature Communications]

Latent transition analysis for longitudinal studies of post-acute infection syndromes

Corresponding Author: Professor Jan Hasenauer

Version 0:

Reviewer comments:

Reviewer #1

(Remarks to the Author)

This paper presents a modelling framework for analysing latent transitions in longitudinal data on post-acute infection syndromes. The manuscript is well written, and the methods are sound.

The authors demonstrate the applicability of their approach using a large multinational real-world dataset. Their findings replicate several well-established observations, such as the higher burden of symptoms among women compared to men, the relative independence of sensory symptoms from other symptom clusters, and the reduced likelihood of post-COVID syndrome (PCS) following infections with later variants of concern. An especially noteworthy contribution is the observation that some patients tend to recover from respiratory PPC, while recovery from severe PCC is rare — a finding that extends beyond existing knowledge.

The proposed modelling framework has some notable strengths:

1. The ability to jointly handle discrete (e.g. presence of symptoms) and continuous features (e.g., health-related quality of life)
2. Efficient estimation of covariate-dependent transitions
3. Implicit handling of missing data

The appealing graphical representations of the results are particularly noteworthy.

I have some specific suggestions:

1. Smoking status, education (or other proxies of socioeconomic status), and body mass index are commonly included as potential predictors in PCS research. From Supplemental Figure 4, it appears that smoking status was considered in the initial model fitting. However, none of these covariates are reported in the results (e.g., Figure 5). The authors should clarify which covariates were initially included but later excluded, and which were deliberately omitted from the analysis.
2. Some comments on the innovative aspects of the proposed approach (e.g., page 4, lines 91–110; page 10, lines 202–205) might be more appropriately placed in the Discussion section.
3. The generalisability of the findings deserves discussion. The cohorts analysed appear to be exclusively hospital-based and therefore exclude individuals with mild or asymptomatic infections who did not seek medical care during the acute phase. States and transitions may differ in population-based cohorts; for instance, more individuals may have started in a “healthy” state, or a “mild acute” state may have been observed.
4. The conclusion states: “In summary, we present an extended LTA framework capable of uncovering the hidden dynamics of the progression of post-acute infection syndromes in a data-driven, unbiased manner.” While the data-driven nature of the approach does protect against certain biases, it is important to acknowledge that the validity of the results still depends on underlying assumptions.

(Remarks on code availability)

I checked the availability of the implementation, which is provided on github and zenodo. However, I did not examine the code in detail.

Reviewer #2

(Remarks to the Author)

The authors present an interesting study based on an important longitudinal database used to investigate post-COVID symptoms. The methodological proposal is relevant, as it extends the hidden Markov model to accommodate both binary and continuous responses, constraining the transition probabilities so that the effects of covariates are estimated only for specific transitions. This type of constraint may also be useful in other analytical contexts. The interpretations and conclusions drawn from the results are clear and further highlight the importance of the proposed model formulation.

The paper is very well written and clear in both its applied and methodological components. The procedure for estimating the model parameters is accurately described, and the software used for computational aspects is made available as open source.

Only one clarification is requested regarding the point indicated below; otherwise, the paper is interesting and can be published in the form presented by the authors.

On page 20, immediately after formulas (1) and (2), the formulation of the adopted model is not entirely clear. Greater precision on this point would be appreciated. Specifically:

- Should the proposed model be interpreted as a multivariate model for mixed response variables?
 - How is the joint probability between the categorical (binary) variable and one or both continuous variables defined exactly?
- A brief clarification on these aspects would make the presentation of the model more transparent and complete.

(Remarks on code availability)

The code is clearly written; I was not able to run it completely, but based on how it is presented, it appears to be readable, correct, and easily usable by other users.

Reviewer #3

(Remarks to the Author)

This manuscript represents an excellent piece of work in which the authors demonstrate how multi-modal data and a large number of latent states and covariates can be incorporated efficiently into latent transition analysis (LTA) to model temporal disease progression both on a population and individual level. The strength of this paper is the combination of a large, high-quality data set from the ORCHESTRA project with a very thorough and partially novel methodological approach for model fitting and selection. A major contribution of the work is the parsimonious parametrization of covariate effects, which allows for the incorporation of covariates into initial states and transition probabilities while keeping the number of parameters that need to be estimated reasonably small. Overall, the approach is generalizable and applicable to other post-acute infectious syndromes (PAISs) for which heterogeneous longitudinal data are available. Reproducing and reusing the results from this work is enabled by the availability of the full implementation of the framework on Github and Zenodo.

The manuscript is well written and to the point, though some aspects of the methodology remain vague and should be explained or commented on before publication.

Detailed comments:

1. The manuscript lacks explanation of how the annotation of the latent states was derived from the estimated emission matrices.
2. Fig. 5A and Supplementary Fig. 11 show a list of 12 different covariates that were included in the 7-state model. In contrast, Supplementary Fig. 4 contains a list of 17 covariates (the 12 from the other two figures plus 5 additional covariates). Why are these two lists different? Which covariates were initially tested for inclusion? Why (or why not) did this list include BMI or cardiovascular disease?
3. Fig. 1 is not referenced in the text. When you introduce this reference, this would be a good place to list the covariates used for the final 7-state model because the caption of Fig. 1 refers to them.
4. line 507: The sentence starting "We note ..." is misleading since neither the observations nor scores are time-independent, but the parameters b_{il} , μ_{il} and σ_{il} of the distribution for their emission probabilities. Please reformulate the sentence. Also, I recommend to mention the term "emission probabilities" in the paragraph starting in line 500.
5. line 509 ff.: Before introducing the notation for beta in line 512, it would be better to first motivate the $N \times (N-1)$ degrees of freedom for the transition probability matrix (due to the N constraints that the row sums are 1) and then explain that each of the $N \times (N-1)$ probabilities consists of a part that is independent of the covariates ($h=1$) and another part that depends on the covariates ($h=2$). Moreover, it would increase clarity to motivate the choice of the logistic regression (I guess to obtain transition probability values between 0 and 1).
6. line 542 ff.: It would improve clarity if the notation for the emission probability $c_i(X_i)$ was moved from the Supplementary (line 873) to the main text.
7. line 596 ff.: For the forward selection approach, does it matter in which order the covariates were added to the model? If it matters, in which order were they added? What does "lowest BIC" (line 600) refer to, i.e. lowest compared to what?

8. Please comment on the compute resources that were needed for model fitting and selection. Was this done on a standard PC/laptop?

Typos:

line 126: delete one of “measured at all times points” or “recorded at each timepoint”

line 254: “the” is missing between “reduces” and “number”

lines 256: change → changes

line 257-258: “...we uncover how...” some word seems to be missing in the remaining sentence

line 494: observation → observations

line 514 “trans” is missing as superscript on rho

line 528: delete one of “using” or “taking”

line 646: “be” is missing between “can” and “found”

SI, equations following line 894, second line: $S_j^t \rightarrow S_j^1$

SI, line 899: The first sentence is incomplete.

SI, line 900: $d \rightarrow D$ (patients)

SI, line 923: The superscripts $t+1$ and 1 need to be deleted from ν since the steady state distribution is time-independent.

SI, line 925: eigenvectors → the first eigenvector, $a \rightarrow A$

(Remarks on code availability)

I followed the instructions in the README file to install the code and I ran the tutorial on my laptop, which went fine. I have not reproduced the results of the paper as this had required better compute resources.

Version 1:

Reviewer comments:

Reviewer #1

(Remarks to the Author)

All my comments have been addressed.

(Remarks on code availability)

Reviewer #2

(Remarks to the Author)

The authors have improved the manuscript during the revision process, and the paper is now suitable for publication.

(Remarks on code availability)

The code provided in Github is well written and well documented. The vignette are clear and explicative. I did not run the code however it seems written with care. The readme file provides all the required instructions.

Reviewer #3

(Remarks to the Author)

The authors carefully addressed all my comments and provided more detailed explanations on the methodology, which increased the clarity of the manuscript. I have no further suggestions for improvement.

(Remarks on code availability)

I already reviewed the code in the first round.

Remarks and Responses to the Reviewers

Reviewer #1

Reviewer #1 (Remarks to the Author):

This paper presents a modelling framework for analysing latent transitions in longitudinal data on post-acute infection syndromes. The manuscript is well written, and the methods are sound.

The authors demonstrate the applicability of their approach using a large multinational real-world dataset. Their findings replicate several well-established observations, such as the higher burden of symptoms among women compared to men, the relative independence of sensory symptoms from other symptom clusters, and the reduced likelihood of post-COVID syndrome (PCS) following infections with later variants of concern. An especially noteworthy contribution is the observation that some patients tend to recover from respiratory PPC, while recovery from severe PCC is rare – a finding that extends beyond existing knowledge.

The proposed modelling framework has some notable strengths:

1. The ability to jointly handle discrete (e.g. presence of symptoms) and continuous features (e.g., health-related quality of life)
2. Efficient estimation of covariate-dependent transitions
3. Implicit handling of missing data

The appealing graphical representations of the results are particularly noteworthy.

Thank you very much for taking the time to thoroughly examine our work, and especially for the positive feedback on the methods, the text and the figures.

I have some specific suggestions:

1. Smoking status, education (or other proxies of socioeconomic status), and body mass index are commonly included as potential predictors in PCS research. From Supplementary Figure 4, it appears that smoking status was considered in the initial model fitting. However, none of these covariates are reported in the results (e.g., Figure 5). The authors should clarify which covariates were initially included but later excluded, and which were deliberately omitted from the analysis.

Thank you for pointing out that the availability and selection of covariates requires clarification. Indeed, Supplementary Figure 4 was misleading in this regard as it highlights all the covariates considered (as well as hospitalisation) in the forward model selection approach. The aim of this figure is to show how the largest number of covariates used across models varying in the number of states, have similar covariate impact patterns. However, considering that these results are not explicitly used to derive the main results of the manuscript, the reader may indeed have difficulty understanding its inclusion.

To address the point, we revised the text. In the revised manuscript, we state the total number of covariates used. Additionally, we now explicitly state the variables which were ultimately selected in the forward selection for the 7 state model, as well as the order in which they appear. This is supported by replacing Supplementary Figure 4 with the enhanced table which displays the order in which the variables were chosen for models with states $N = 4$ to $N = 8$. Together,

we hope this elucidates the list of patient covariates which were considered by the forward model selection approach and the covariate list ultimately chosen for the 7 state model.

Line 164:

Initial Submission:

The assessment of the models with 4 to 8 latent states revealed a high degree of structural similarity (Supplementary Figure 3), as well as similar covariate impact (Supplementary Figure 4). Model selection reveals...

Revised Submission:

The assessment of the models with 4 to 8 latent states revealed a high degree of structural similarity (Supplementary Figure 3), ~~as well as similar covariate impact (Supplementary Figure 4)~~. Model selection reveals...

Lines 631:

Initial Submission:

Model selection was completed by first estimating the null HMM model for a given number of latent states, without the influence of covariates in the initial and transition state probabilities, using the 2-stage optimisation approach with 20 multistarts each. 20 starts were chosen due to available computation resources but were seen to be sufficient in finding similar solutions across starts. Age and sex covariates are then added to describe the initial and transition probabilities and the parameters are re-estimated with a warm start using the estimated parameters for the best fitting null model for each number of states, N . A forward selection approach is then made, where a single covariate is introduced to the age/sex model, and a similar re-estimation is completed using a warm start. The corresponding Bayesian Information Criterion (BIC) value [44] is calculated after estimating the new covariate model, and the covariate is selected and added to the base model if it returns the lowest BIC value. This process is repeated until an improvement less than 10 occurs across all covariates. This was completed for 5 models (4 to 8 latent states) and we chose the final model using the BIC as there is strong support for its usage in identifying the most suitable number of latent states [45-47]. Hospitalisation was excluded from the selection process as it is strongly correlated with unrecorded medical complications, making the clinical meaning difficult to interpret. In comparing models varying in size from $N = 4$ to 8 states, it was found that the $N = 7$ model gave the smallest BIC value in comparison to both the null (no covariates used) and forward-selected model groups (Supplementary Figure 2). To evaluate model performance of the final 7 state model, the chosen set of covariates was used in a 5 fold cross validation procedure with 20 multistarts, each with a cold starting parameter vector and not utilising the 2-stage approach.

Revised Submission:

Model selection was completed by first estimating the null HMM model for a given number of latent states, without the influence of covariates in the initial and transition state probabilities, using the 2-stage optimisation approach with 20 multistarts each. 20 starts were chosen due to available computation resources but were seen to be sufficient in finding similar solutions across starts. Age and sex covariates are then added to describe the initial and transition probabilities and the parameters are re-estimated with a warm start using the estimated parameters for the best fitting null model for each number of states, N . A forward selection approach is then made, where a single

covariate is introduced to the age/sex model, and a similar re-estimation is completed using a warm start. The corresponding Bayesian Information Criterion (BIC) value [44] is calculated after estimating the new covariate model, and the covariate is selected and added to the base model if it returns the lowest BIC value **amongst all covariates which are currently not included in the model at the selection round**. This process is repeated until an improvement less than 10 occurs across all covariates. This was completed for 5 models (4 to 8 latent states) and we chose the final model using the BIC as there is strong support for its usage in identifying the most suitable number of latent states [45-47]. ~~Hospitalisation was excluded from the selection process as it is strongly correlated with unrecorded medical complications, making the clinical meaning difficult to interpret.~~

In comparing models varying in size from $N = 4$ to 8 states **using the forward selection**, it was found that the $N = 7$ model gave the **smallest lowest** BIC value in comparison to both the null (no covariates used) and forward-selected model groups (Supplementary Figure 2). **For the 7 state model, the following covariates were selected in the order mentioned: oxygen therapy, infection wave, chronic respiratory disease, and corticosteroids (Supplementary Figure 4).** These covariates were added consecutively to the base model that included age and sex already, which resulted in the final 7 state model presented. A patient's history of smoking as well as therapy history at the acute stage (usage of antivirals (remdesivir), monoclonal antibodies, immunomodulators and vaccination before the acute infection) were also considered in the forward selection approach but were ultimately not selected. Hospitalisation was excluded from the list of possible covariates as it is strongly correlated with unrecorded medical complications, **making the clinical meaning difficult to interpret.** To evaluate model performance of the final 7 state model, the chosen set of covariates was used in a 5 fold cross validation procedure with 20 multistarts, each with a cold starting parameter vector and not utilising the 2-stage approach.

Supplementary Figure 4:

Initial Submission:

Scatter plot of parameter estimates of $\bar{\rho}^{initial}$ and $\bar{\rho}^{trans}$ for 20 multistarts using all

covariates. The influence of covariate is consistent across varying number of states (N). More blue lines indicates parameters with a larger likelihood value.

Revised Submission:

Forward Selection Process across Model Sizes (4 to 8 States)

Covariate	N = 4	N = 5	N = 6	N = 7	N = 8
Hospital Admission					
Age	0	0	0	0	0
Sex	0	0	0	0	0
Infection Wave	2	2	1	2	
Chronic Respiratory Disease	1	1		3	2
Oxygen Therapy	3			1	3
Corticosteroids		4	2	4	
Monoclonal Antibodies		3			1
Immunomodulators		5			
Previous Smoker					
Antivirals (Remdesivir)					
Vaccination before Acute Infection					
Model BIC	59784.15	57741.57	56670.70	56276.02	56295.32
Difference between Model BIC and N = 7 BIC (Δ BIC)	+3508.13	+1465.55	+394.68	+0.00	+19.30

Selection of Covariates across Model Sizes. The enhanced table displays which covariates were selected during the forward model selection process for model sizes $N = 4$ to $N = 8$. The integer value within the cell indicates at which selection round was the covariates chosen, with 1 being the first covariate chosen. A grey strike indicates the covariate was not chosen. Hospitalisation was excluded from the list of possible covariates, while age and sex were always included in every model. The 7 state model was found to have the lowest BIC value.

2. Some comments on the innovative aspects of the proposed approach (e.g., page 4, lines 91-110; page 10, lines 202-205) might be more appropriately placed in the Discussion section.

We thank the reviewer for pointing out that the section rather reads more as a part of the discussion/conclusion. To address the issue, we rephrased large parts of the section using more descriptive formulations. Additionally, page 10 lines 202-205 have been moved and rephrased in the discussion section where it is more appropriate. In the revised version, the interpretation of the results should be only based in the Discussion section.

Lines 91:

Initial Submission:

To provide a comprehensive and scalable assessment of PAISs, we propose a flexible and interpretable LTA framework. The framework builds on established Hidden Markov Modelling implementations [18, 19], addressing the specific challenges of longitudinal cohort studies with heterogeneous data types (i.e., binary and continuous), as well as incomplete observations that arise over long follow-up periods. A key innovation of our approach

lies in a parsimonious parameterisation of the covariate-dependent transition structure. Rather than modelling the full matrix of covariate effects separately for each transition probability, which would lead to an unmanageable number of parameters, patient characteristics are projected onto a low-dimensional scalar representation that modulates the entire transition matrix in an interpretable and computationally efficient way. This significantly reduces overfitting risk while preserving individual-level heterogeneity. We conducted a comprehensive simulation-based validation study to evaluate the robustness of parameter recovery, predictive performance, and covariate interpretability across a range of controlled scenarios. These experiments confirm the framework’s reliability under varying data sparsity, symptom noise, and covariate effects, providing confidence in its application to real-world longitudinal data.

In addition to characterising state transitions, our framework supports state filtering for individual patients, making use of prior symptom and HRQoL history to improve future symptom and HRQoL predictions. By applying a recursive state update procedure, we dynamically refine patient-level latent state probabilities at each timepoint and generate predictions for both binary and continuous variables.

Full methodological details, including model specification, estimation procedures, and evaluation metrics, are provided in the Methods section. The full implementation of the framework is provided, including model fitting, prediction, and visualisation routines, available on Github  and archived on Zenodo .

Revised Submission:

To provide a comprehensive and scalable assessment of PAISs, we propose a flexible and interpretable LTA framework (see Figure 1 for a visual outline). The framework builds on established Hidden Markov Modelling implementations [18, 19], addressing the specific challenges of longitudinal cohort studies with heterogeneous data types (i.e., binary and continuous), as well as incomplete observations that arise over long follow-up periods.

~~A key innovation of our approach lies in a parsimonious parameterisation of the covariate-dependent transition structure. Rather than modelling the full matrix of covariate effects separately for each transition probability, which would lead to an unmanageable number of parameters, patient characteristics are projected onto a low-dimensional scalar representation that modulates the entire transition matrix in an interpretable and computationally efficient way. This significantly reduces overfitting risk while preserving individual-level heterogeneity. We conducted a comprehensive simulation-based validation study to evaluate the robustness of parameter recovery, predictive performance, and covariate interpretability across a range of controlled scenarios. These experiments confirm the framework’s reliability under varying data sparsity, symptom noise, and covariate effects, providing confidence in its application to real-world longitudinal data.~~

In order to circumvent the large number of parameters present when modelling the full matrix of covariate effects separately for each transition probability, patient characteristics are projected onto a low-dimensional scalar representation that modulates the entire transition matrix in an interpretable and computationally efficient way.

We conducted a comprehensive simulation-based validation study to evaluate the robustness of parameter recovery, predictive performance, and covariate interpretability across a range of controlled scenarios. These experiments confirm the framework’s reliability under varying data sparsity, symptom noise, and covariate effects, providing confidence in its application to real-world longitudinal data. Indeed, we found our approach to significantly reduce overfitting risk while preserving individual-level heterogeneity.

In addition to characterising state transitions, our framework supports state filtering for individual patients, making use of prior symptom and HRQoL history to improve future symptom and HRQoL predictions (Figure 1f). ~~By applying a recursive state update procedure, we dynamically refine patient-level latent state probabilities at each timepoint and generate predictions for both binary and continuous variables.~~

Full methodological details, including model specification, estimation procedures, and evaluation metrics, are provided in the Methods section. The full implementation of the framework is provided, including model fitting, prediction, and visualisation routines, available on Github  and archived on Zenodo .

Lines 217:

Initial Submission:

Our analysis revealed that LTA based on flexible HMMs allows for the data-driven definition of Acute Infection and PCC states. Model selection allows for robust assessment of the number of distinct states, and subsequent model simulations can be used to identify transition patterns between states.

Revised Submission:

~~Our analysis revealed that LTA based on flexible HMMs allows for the data-driven definition of Acute Infection and PCC states. Model selection allows for robust assessment of the number of distinct states, and subsequent model simulations can be used to identify transition patterns between states.~~

Line 384:

Initial Submission:

...demonstrating the plausibility of our modelling approach.

Our analysis confirms known features of PCC while offering novel insights...

Revised Submission:

...demonstrating the plausibility of our modelling approach.

~~Our analysis revealed that LTA based on flexible HMMs allows for the data-driven definition of Acute Infection and PCC states. Model selection allows for robust assessment of the number of distinct states, and subsequent model simulations can be used to identify transition patterns between states.~~

Our analysis **also** confirms known features of PCC while offering novel insights...

3. The generalisability of the findings deserves discussion. The cohorts analysed appear to be exclusively hospital-based and therefore exclude individuals with mild or asymptomatic infections who did not seek medical care during the acute phase. States and transitions may differ in population-based cohorts; for instance, more individuals may have started in a “healthy” state, or a “mild acute” state may have been observed.

We thank the reviewer for pointing out that the composition of the cohort, as well as the

resulting limitations were not properly discussed. Indeed, the cohort contains 3824 individuals (75.4%) who were hospitalized, and 1254 individuals (24.6%) which were not admitted to hospital during the acute stage (Supplementary Table 7). This ratio is indeed not representative for individuals suffering from Long Covid.

Although the dataset is not completely balanced across the hospitalisation variable, we did utilise more than 1000 patients, composing nearly 1/4 of the data used for the analysis, which does provide decent statistical power to comment on the personalised Long COVID trajectory of the general population, including hospitalised and non-hospitalised patients. When utilising the characteristics of the ORCHESTRA dataset, which are reflective of a majority-hospitalised group (such as the frequent trajectories in Figure 5c), then the result may not necessarily extend to non-majority hospitalised groups, such as the general population. We have now stressed this at the very first point to our limitations section.

Line 418:

Initial Submission:

Despite its strengths, our study is subject to several limitations. Firstly, there are data collection constraints due to pandemic stress on health systems. In addition, patient attrition at...

Revised Submission:

Despite its strengths, our study is subject to several limitations. Firstly, there are data collection constraints due to pandemic stress on health systems. **Importantly, the majority of patients involved in our study were either admitted to general or ICU hospital wards at the acute stage of infection (75.4%), which is not representative of the general European population who were hospitalised from SARS-CoV-2 [28]. This implies that when utilising the model in conjunction with patients and their characteristics of the ORCHESTRA cohort, there may be a smaller number of trajectories ending in the Healthy state than when applying to the general population.** In addition, patient attrition at...

4. The conclusion states: “In summary, we present an extended LTA framework capable of uncovering the hidden dynamics of the progression of post-acute infection syndromes in a data-driven, unbiased manner.” While the data-driven nature of the approach does protect against certain biases, it is important to acknowledge that the validity of the results still depends on underlying assumptions.

This is a fair point, and the use of the term “unbiased” was too liberal. In the sentence mentioned, we believe “data-driven” is the more accurate term to describe the methodology presented, so “unbiased” is removed.

Line 469:

Initial Submission:

In summary, we present an extended LTA framework capable of uncovering the hidden dynamics of the progression of post-acute infection syndromes in a data-driven, unbiased manner.

Revised Submission:

In summary, we present an extended LTA framework capable of uncovering the hidden dynamics of the progression of post-acute infection syndromes in a data-driven, ~~unbiased~~ manner.

In addition, in the preceding paragraph, we also feel the term should be replaced to describe the meaning more specifically.

Line 454:

Initial Submission:

LTA constitutes an unbiased, data-driven method that allows health states to be inferred directly from observed data, that can robustly characterise disease heterogeneity and capture the evolving dynamics of long-term conditions.

Revised Submission:

LTA constitutes ~~an unbiased~~, data-driven method that allows health states to be inferred directly from observed data **while relying on few underlying model assumptions**, that can robustly characterise disease heterogeneity and capture the evolving dynamics of long-term conditions.

Reviewer 1 (Remarks on code availability):

I checked the availability of the implementation, which is provided on github and zenodo. However, I did not examine the code in detail.

We thank for reviewer for confirming the code availability.

Reviewer #2

Reviewer #2 (Remarks to the Author):

The authors present an interesting study based on an important longitudinal database used to investigate post-COVID symptoms. The methodological proposal is relevant, as it extends the hidden Markov model to accommodate both binary and continuous responses, constraining the transition probabilities so that the effects of covariates are estimated only for specific transitions. This type of constraint may also be useful in other analytical contexts. The interpretations and conclusions drawn from the results are clear and further highlight the importance of the proposed model formulation.

The paper is very well written and clear in both its applied and methodological components. The procedure for estimating the model parameters is accurately described, and the software used for computational aspects is made available as open source.

Many thanks for the kind words and thoughtful perspective on how our approach can be relevant in other applications. We really appreciate the time taken for you to review our work.

Only one clarification is requested regarding the point indicated below; otherwise, the paper is interesting and can be published in the form presented by the authors. On page 20, immediately after formulas (1) and (2), the formulation of the adopted model is not entirely clear. Greater precision on this point would be appreciated. Specifically:

- Should the proposed model be interpreted as a multivariate model for mixed response variables?
- How is the joint probability between the categorical (binary) variable and one or both continuous variables defined exactly?

A brief clarification on these aspects would make the presentation of the model more transparent and complete.

We thank the reviewer for pointing out that the mathematical details of the model and its interpretation is not described sufficiently well. We tried to clarify the two points in the revised manuscript. In brief:

Question 1: Yes, ultimately for a given collection of covariates, we have a corresponding set of predicted mixed (binary and continuous) responses or observational outcomes at a given timepoint. But as the transition between states is governed by a Markov process, the response variables at the different time points are correlated. We have now added clarifying statements which we hope guides the reader to a better intuition of the model.

Question 2: The definition of the joint probability is related to the definition of c which is located in the Supplementary, but has now been brought into the main text to improve clarity. The revised text now includes the expression of the probability of observing a binary symptom/HRQoL score, as well as a reaffirmation of the assumptions of the model. We hope this rephrasing makes it clear that how the joint probability is obtained.

Line 533:

Initial Submission:

We note that both the binary symptom observations and continuous HRQoL scores are independent of time.

Revised Submission:

~~We note that both the binary symptom observations and continuous HRQoL scores are independent of time.~~ We note that the parameters defining the emissions (b_{il} , μ_{il} and σ_{il}) are independent of time, whereas the binary symptom observations and continuous HRQoL scores are dependent. This is because observations themselves only depend on the current state, where the state probability changes from one timepoint to the next. The probability of an observation X_l^t for an individual in the i -th state is defined as

$$P(X_l^t | S_i^t = 1) \equiv c_i(X_l).$$

Assuming conditional independence among observations at a specific time point, the joint probability of a set of observations at this time point is obtained as the product of their individual probabilities, conditioned on the occurrence of the i -th state.

Line 564:

Initial Submission:

A similar parametrisation is used for the initial state distribution with $r^{initial} = \vec{\rho}^{initial} \cdot \vec{C}^{initial}$, which calculates the probability of being in the i -th state at the first timepoint, $\pi_i = P(S_i^{t=0})$, using $i = 1$ as the reference state (Supplementary 1.1.1).

Revised Submission:

A similar parametrisation is used for the initial state distribution with $r^{initial} = \vec{\rho}^{initial} \cdot \vec{C}^{initial}$, which calculates the probability of being in the i -th state at the first timepoint, $\pi_i = P(S_i^{t=0})$, using $i = 1$ as the reference state (Supplementary 1.1.1). **Thus, a patient's covariates affect the probability of state occurrences at each timepoint, which in turn determines how likely the patient exhibits a given symptom or HRQoL score.**

Reviewer #2 (Remarks on code availability):

The code is clearly written; I was not able to run it completely, but based on how it is presented, it appears to be readable, correct, and easily usable by other users.

We thank the reviewer for confirming the code availability and for assessing the code quality.

Reviewer #3

Reviewer #3 (Remarks to the Author):

This manuscript represents an excellent piece of work in which the authors demonstrate how multi-modal data and a large number of latent states and covariates can be incorporated efficiently into latent transition analysis (LTA) to model temporal disease progression both on a population and individual level. The strength of this paper is the combination of a large, high-quality data set from the ORCHESTRA project with a very thorough and partially novel methodological approach for model fitting and selection. A major contribution of the work is the parsimonious parametrization of covariate effects, which allows for the incorporation of covariates into initial states and transition probabilities while keeping the number of parameters that need to be estimated reasonably small. Overall, the approach is generalizable and applicable to other post-acute infectious syndromes (PAISs) for which heterogeneous longitudinal data are available. Reproducing and reusing the results from this work is enabled by the availability of the full implementation of the framework on Github and Zenodo. The manuscript is well written and to the point, though some aspects of the methodology remain vague and should be explained or commented on before publication.

Thank you for the high praise of our work and your thorough perspective on what can be improved on the manuscript. We feel that your review of the methodology strongly helps clarify the more technical sections and we very much appreciate this feedback.

Detailed comments:

- The manuscript lacks explanation of how the annotation of the latent states was derived from the estimated emission matrices.

We thank the reviewer for pointing out that additional details on the annotation process would be valuable. In brief, the labelling was conducted in collaboration with clinical experts based on the available definitions of long-COVID states, which was not stated in the initial submission. We have added a paragraph to the third results section that explicitly states how labelling of states was conducted now.

Line 159:

Initial Submission:

-

Revised Submission:

By inspecting the emission probabilities, states were interpreted as representing distinct health profiles, where the labelling process involved close examination of the emission patterns by clinical experts to identify the clinical meaning of the state. For example, a state with very low probabilities of manifesting any symptom and high probability of high HRQoL scores can be clearly annotated as a healthy state.

- Fig. 5A and Supplementary Fig. 11 show a list of 12 different covariates that were included in the 7-state model. In contrast, Supplementary Fig. 4 contains a list of 17 covariates (the 12 from the other two figures plus 5 additional covariates). Why are these two lists different? Which covariates were initially tested for inclusion? Why (or why not) did this list include BMI or cardiovascular disease?

This is an important clarification to make which was also brought up by Reviewer #1 in their first comment. Thank you for raising it. The purpose of Supplementary Figure 4 was to highlight the structure similarity of the covariate impact for various models sizes using all covariates available. Yet, this does generate confusion in how the forward model selection process was conducted.

We have addressed this point in the text by adding clarifying statements to several section in order to make clear how covariates were ultimately selected. This includes revised text in the model selection section, as well as replacing Supplementary Figure 4 with a new enhanced table figure which specifies the order in which the covariates were added in the model selection process.

Line 164:

Initial Submission:

The assessment of the models with 4 to 8 latent states revealed a high degree of structural similarity (Supplementary Figure 3), as well as similar covariate impact (Supplementary Figure 4). Model selection reveals...

Revised Submission:

The assessment of the models with 4 to 8 latent states revealed a high degree of structural similarity (Supplementary Figure 3), ~~as well as similar covariate impact (Supplementary Figure 4)~~. Model selection reveals...

Line 631:

Initial Submission:

Model selection was completed by first estimating the null HMM model for a given number of latent states, without the influence of covariates in the initial and transition state probabilities, using the 2-stage optimisation approach with 20 multistarts each. 20 starts were chosen due to available computation resources but were seen to be sufficient in finding similar solutions across starts. Age and sex covariates are then added to describe the initial and transition probabilities and the parameters are re-estimated with a warm start using the estimated parameters for the best fitting null model for each number of states, N . A forward selection approach is then made, where a single covariate is introduced to the age/sex model, and a similar re-estimation is completed using a warm start. The corresponding Bayesian Information Criterion (BIC) value [44] is calculated after estimating the new covariate model, and the covariate is selected and added to the base model if it returns the lowest BIC value. This process is repeated until an improvement less than 10 occurs across all covariates. This was completed for 5 models (4 to 8 latent states) and we chose the final model using the BIC as there is strong support for its usage in identifying the most suitable number of latent states [45-47]. Hospitalisation was excluded from the selection process as it is strongly correlated with unrecorded medical complications, making the clinical meaning difficult to interpret. In comparing models varying in size from $N = 4$ to 8 states, it was found that the $N = 7$ model gave the smallest BIC value in comparison to both the null (no covariates used) and forward-selected model groups (Supplementary Figure 2). To evaluate model performance of the final 7 state model, the chosen set of covariates was used in a 5 fold cross validation procedure with 20 multistarts, each with a cold starting parameter vector and not utilising the 2-stage approach.

Revised Submission:

Model selection was completed by first estimating the null HMM model for a given number of latent states, without the influence of covariates in the initial and transition state probabilities, using the 2-stage optimisation approach with 20 multistarts each. 20 starts were chosen due to available computation resources but were seen to be sufficient in finding similar solutions across starts. Age and sex covariates are then added to describe the initial and transition probabilities and the parameters are re-estimated with a warm start using the estimated parameters for the best fitting null model for each number of states, N . A forward selection approach is then made, where a single covariate is introduced to the age/sex model, and a similar re-estimation is completed using a warm start. The corresponding Bayesian Information Criterion (BIC) value [44] is calculated after estimating the new covariate model, and the covariate is selected and added to the base model if it returns the lowest BIC value **amongst all covariates which are currently not included in the model at the selection round**. This process is repeated until an improvement less than 10 occurs across all covariates. This was completed for 5 models (4 to 8 latent states) and we chose the final model using the BIC as there is strong support for its usage in identifying the most suitable number of latent states [45-47]. ~~Hospitalisation was excluded from the selection process as it is strongly correlated with unrecorded medical complications, making the clinical meaning difficult to interpret.~~

In comparing models varying in size from $N = 4$ to 8 states **using the forward selection**, it was found that the $N = 7$ model gave the **smallestlowest** BIC value in comparison to both the null (no covariates used) and forward-selected model groups (Supplementary Figure 2). **For the 7 state model, the following covariates were selected in the order mentioned: oxygen therapy, infection wave, chronic respiratory disease, and corticosteroids (Supplementary Figure 4).** These covariates were added consecutively to the base model that included age and sex already, which resulted in the final 7 state model presented. A patient's history of smoking as well as therapy history at the acute stage (usage of antivirals (remdesivir), monoclonal antibodies, immunomodulators and vaccination before the acute infection) were also considered in the forward selection approach but were ultimately not selected. Hospitalisation was excluded from the list of possible covariates as it is strongly correlated with unrecorded medical complications, **making the clinical meaning difficult to interpret**. To evaluate model performance of the final 7 state model, the chosen set of covariates was used in a 5 fold cross validation procedure with 20 multistarts, each with a cold starting parameter vector and not utilising the 2-stage approach.

Supplementary Figure 4:

Initial Submission:

Scatter plot of parameter estimates of $\hat{\rho}^{initial}$ and $\hat{\rho}^{trans}$ for 20 multistarts using all covariates. The influence of covariate is consistent across varying number of states (N). More blue lines indicates parameters with a larger likelihood value.

Revised Submission:

Forward Selection Process across Model Sizes (4 to 8 States)

Covariate	N = 4	N = 5	N = 6	N = 7	N = 8
Hospital Admission					
Age	0	0	0	0	0
Sex	0	0	0	0	0
Infection Wave	2	2	1	2	
Chronic Respiratory Disease	1	1		3	2
Oxygen Therapy	3			1	3
Corticosteroids		4	2	4	
Monoclonal Antibodies		3			1
Immunomodulators		5			
Previous Smoker					
Antivirals (Remdesivir)					
Vaccination before Acute Infection					
Model BIC	59784.15	57741.57	56670.70	56276.02	56295.32
Difference between Model BIC and N = 7 BIC (Δ BIC)	+3508.13	+1465.55	+394.68	+0.00	+19.30

Selection of Covariates across Model Sizes. The enhanced table displays which covariates were selected during the forward model selection process for model sizes $N = 4$ to $N = 8$. The integer value within the cell indicates at which selection round was the covariates chosen, with 1 being the first covariate chosen. A grey strike indicates the covariate was not chosen. Hospitalisation was excluded from the list of possible covariates, while age and sex were always included in every model. The 7 state model was found to have the lowest BIC value.

- Fig. 1 is not referenced in the text. When you introduce this reference, this would be a good place to list the covariates used for the final 7-state model because the caption of Fig. 1 refers to them.

Thank you for pointing this out. References to the figure have been added in the first section of the results now, as well as a revising of some of the formulation of the section. Additionally we have added significantly more detail to the manuscript on the topic of how the covariates were dealt with. However, we felt these changes would be more aptly placed in the methodology section as it contains more details in how the final model was arrived at. Please see the reply to the previous comment for specific details.

Line 91:

Initial Submission:

To provide a comprehensive and scalable assessment of PAISs, we propose a flexible and interpretable LTA framework. The framework builds on established Hidden Markov Modelling implementations [18, 19], addressing the specific challenges of longitudinal cohort studies with heterogeneous data types (i.e., binary and continuous), as well as incomplete observations that arise over long follow-up periods. A key innovation of our approach lies in a parsimonious parameterisation of the covariate-dependent transition structure. Rather than modelling the full matrix of covariate effects separately for each transition probability, which would lead to an unmanageable number of parameters, patient characteristics are projected onto a low-dimensional scalar representation that modulates the entire transition matrix in an interpretable and computationally efficient way. This significantly reduces overfitting risk while preserving individual-level heterogeneity. We conducted a comprehensive simulation-based validation study to evaluate the robustness of parameter recovery, predictive performance, and covariate interpretability across a range of controlled scenarios. These experiments confirm the framework’s reliability under varying data sparsity, symptom noise, and covariate effects, providing confidence in its application to real-world longitudinal data.

In addition to characterising state transitions, our framework supports state filtering for individual patients, making use of prior symptom and HRQoL history to improve future symptom and HRQoL predictions. By applying a recursive state update procedure, we dynamically refine patient-level latent state probabilities at each timepoint and generate predictions for both binary and continuous variables.

Full methodological details, including model specification, estimation procedures, and evaluation metrics, are provided in the Methods section. The full implementation of the framework is provided, including model fitting, prediction, and visualisation routines, available on Github  and archived on Zenodo .

Revised Submission:

To provide a comprehensive and scalable assessment of PAISs, we propose a flexible and interpretable LTA framework (see Figure 1 for a visual outline). The framework builds on established Hidden Markov Modelling implementations [18, 19], addressing the specific challenges of longitudinal cohort studies with heterogeneous data types (i.e., binary and continuous), as well as incomplete observations that arise over long follow-up periods.

~~A key innovation of our approach lies in a parsimonious parameterisation of the covariate-dependent transition structure. Rather than modelling the full matrix of~~

~~covariate effects separately for each transition probability, which would lead to an unmanageable number of parameters, patient characteristics are projected onto a low-dimensional scalar representation that modulates the entire transition matrix in an interpretable and computationally efficient way. This significantly reduces overfitting risk while preserving individual-level heterogeneity. We conducted a comprehensive simulation-based validation study to evaluate the robustness of parameter recovery, predictive performance, and covariate interpretability across a range of controlled scenarios. These experiments confirm the framework’s reliability under varying data sparsity, symptom noise, and covariate effects, providing confidence in its application to real-world longitudinal data.~~

In order to circumvent the large number of parameters present when modelling the full matrix of covariate effects separately for each transition probability, patient characteristics are projected onto a low-dimensional scalar representation that modulates the entire transition matrix in an interpretable and computationally efficient way.

We conducted a comprehensive simulation-based validation study to evaluate the robustness of parameter recovery, predictive performance, and covariate interpretability across a range of controlled scenarios. These experiments confirm the framework’s reliability under varying data sparsity, symptom noise, and covariate effects, providing confidence in its application to real-world longitudinal data. Indeed, we found our approach to significantly reduce overfitting risk while preserving individual-level heterogeneity.

In addition to characterising state transitions, our framework supports state filtering for individual patients, making use of prior symptom and HRQoL history to improve future symptom and HRQoL predictions (Figure 1f). ~~By applying a recursive state update procedure, we dynamically refine patient-level latent state probabilities at each timepoint and generate predictions for both binary and continuous variables.~~

Full methodological details, including model specification, estimation procedures, and evaluation metrics, are provided in the Methods section. The full implementation of the framework is provided, including model fitting, prediction, and visualisation routines, available on Github  and archived on Zenodo .

- line 507: The sentence starting “We note ...” is misleading since neither the observations nor scores are time-independent, but the parameters b_{il} , μ_{il} and σ_{il} of the distribution for their emission probabilities. Please reformulate the sentence.

This is true and we have now revised the paragraph to incorporate multiple reviewer feedback. This includes reformulating which terms are indeed time-independent, as well as bring the definition of c into the main text, and an explanation of the joint probability of observations.

Line 533:

Initial Submission:

We note that both the binary symptom observations and continuous HRQoL scores are independent of time.

Revised Submission:

~~We note that both the binary symptom observations and continuous HRQoL scores are independent of time.~~We note that the parameters defining the emissions (b_{il} , μ_{il} and σ_{il}) are independent of time, whereas the binary symptom observations and continuous

HRQoL scores are dependent. This is because observations themselves only depend on the current state, where the state probability changes from one timepoint to the next. The probability of an observation X_i^t for an individual in the i -th state is defined as

$$P(X_i^t | S_i^t = 1) \equiv c_i(X_i^t).$$

Assuming conditional independence among observations at a specific time point, the joint probability of a set of observations at this time point is obtained as the product of their individual probabilities, conditioned on the occurrence of the i -th state.

Also, I recommend to mention the term “emission probabilities” in the paragraph starting in line 500.

The term has been mentioned according to the reviewer’s advice.

Line 526:

Initial Submission:

The i -th state is responsible for a particular probability, b_{il} of manifesting the l -th discrete observed symptom.

Revised Submission:

The i -th state is responsible for a particular probability, b_{il} of manifesting the l -th discrete observed symptom-, which are collectively define as the emission probabilities of the binary symptom observations.

- line 509 ff.: Before introducing the notation for beta in line 512, it would be better to first motivate the $N \times (N-1)$ degrees of freedom for the transition probability matrix (due to the N constraints that the row sums are 1) and then explain that each of the $N \times (N-1)$ probabilities consists of a part that is independent of the covariates ($h=1$) and another part that depends on the covariates ($h=2$). Moreover, it would increase clarity to motivate the choice of the logistic regression (I guess to obtain transition probability values between 0 and 1).

Thanks for the suggestion. The revised text now additionally describes the appropriate degrees for the transition probabilities, as well as the choice of the multinomial logistic regression function for parametrisation. We hope the new text added provides a better intuition into how the transition rates are constructed now, as well as why there are $2 \times N \times (N - 1)$ number of parameters.

Line 542:

Initial Submission:

As the severity state of a patient can change within the follow-up time period, there is an associated transition rate, a_{ij} , which is the probability of moving from the i -th state to the j -th state, dependant on a patient’s associated covariates, \vec{C}^{trans} . These transition probabilities are parameterised by a multinomial logistic regression function with regression coefficients $\beta_{hij}^{trans} \in \mathbb{R}^{2 \times N \times (N-1)}$. In order to capture...

Revised Submission:

As the severity state of a patient can change within the follow-up time period, there is an associated transition rate, a_{ij} , which is the probability of moving from the i -th state to the j -th state, dependant on a patient's associated covariates, \vec{C}^{trans} . The matrix a_{ij} is row-stochastic, and so the i -th row has $N - 1$ degrees of freedom as it must sum to 1, and there are total N rows. To ensure that these transition probabilities are bounded between 0 and 1 while being dependant on a patient covariates, we parametrise each row of a_{ij} by a multinomial logistic regression function. Thus, the logistic function parametrising the transition probabilities has regression coefficients $\beta_{hij}^{trans} \in \mathbb{R}^{2 \times N \times (N-1)}$, where the function consists of 2 components; the intercept β_{1ij} and the coefficient, β_{2ij} used in conjunction with the covariate term. ~~These transition probabilities are parameterised by a multinomial logistic regression function with regression coefficients $\beta_{hij}^{trans} \in \mathbb{R}^{2 \times N \times (N-1)}$.~~ In order to capture...

- line 542 ff.: It would improve clarity if the notation for the emission probability $c_i(X_i)$ was moved from the Supplementary (line 873) to the main text.

Thank you for noting this. As mentioned in an earlier comment, we felt it necessary to rework this paragraph entirely, incorporating the material from the Supplementary section as per your suggestion. This includes explicitly stating which terms are time independent, the explicit definition of c , as well as characterising the joint probability of observations.

Line 533:

Initial Submission:

We note that both the binary symptom observations and continuous HRQoL scores are independent of time.

Revised Submission:

~~We note that both the binary symptom observations and continuous HRQoL scores are independent of time.~~ We note that the parameters defining the emissions (b_{il} , μ_{il} and σ_{il}) are independent of time, whereas the binary symptom observations and continuous HRQoL scores are dependent. This is because observations themselves only depend on the current state, where the state probability changes from one timepoint to the next. The probability of an observation X_i^t for an individual in the i -th state is defined as

$$P(X_i^t | S_i^t = 1) \equiv c_i(X_i^t).$$

Assuming conditional independence among observations at a specific time point, the joint probability of a set of observations at this time point is obtained as the product of their individual probabilities, conditioned on the occurrence of the i -th state.

- line 596 ff.: For the forward selection approach, does it matter in which order the covariates were added to the model? If it matters, in which order were they added? What does “lowest BIC” (line 600) refer to, i.e. lowest compared to what?

The ordering of the added covariates does not matter insofar that final 7-state model reached

utilises the set of covariates found during the forward selection process. However, depending on the ordering of the adding of the variables, a different covariate could be chosen at a given selection round if the preceding covariates chosen are altered and the corresponding BIC is reached is different. The resulting final model may then have a different BIC score. However, considering at each selection round, the covariate that gives the best model improvement (smallest BIC score) is chosen, the model selection process is consistently reproducible. Ultimately, a search of all possible combinations of covariates of the model would be needed to identify the smallest BIC model with certainty, but is computationally infeasible.

We now write the order in which the covariates were added to highlight how the forward selection procedure was conducted, with Supplementary Figure 4 being replaced with a new enhanced table added to the manuscript to support the statement. As this is related to previous comments, the section has been revised too. Clarifying statements to the model selection process have been added, and the explicit order in which covariates are added, has also been introduced into the text now.

Line 631:

Initial Submission:

Model selection was completed by first estimating the null HMM model for a given number of latent states, without the influence of covariates in the initial and transition state probabilities, using the 2-stage optimisation approach with 20 multistarts each. 20 starts were chosen due to available computation resources but were seen to be sufficient in finding similar solutions across starts. Age and sex covariates are then added to describe the initial and transition probabilities and the parameters are re-estimated with a warm start using the estimated parameters for the best fitting null model for each number of states, N . A forward selection approach is then made, where a single covariate is introduced to the age/sex model, and a similar re-estimation is completed using a warm start. The corresponding Bayesian Information Criterion (BIC) value [44] is calculated after estimating the new covariate model, and the covariate is selected and added to the base model if it returns the lowest BIC value. This process is repeated until an improvement less than 10 occurs across all covariates. This was completed for 5 models (4 to 8 latent states) and we chose the final model using the BIC as there is strong support for its usage in identifying the most suitable number of latent states [45-47]. Hospitalisation was excluded from the selection process as it is strongly correlated with unrecorded medical complications, making the clinical meaning difficult to interpret. In comparing models varying in size from $N = 4$ to 8 states, it was found that the $N = 7$ model gave the smallest BIC value in comparison to both the null (no covariates used) and forward-selected model groups (Supplementary Figure 2). To evaluate model performance of the final 7 state model, the chosen set of covariates was used in a 5 fold cross validation procedure with 20 multistarts, each with a cold starting parameter vector and not utilising the 2-stage approach.

Revised Submission:

Model selection was completed by first estimating the null HMM model for a given number of latent states, without the influence of covariates in the initial and transition state probabilities, using the 2-stage optimisation approach with 20 multistarts each. 20 starts were chosen due to available computation resources but were seen to be sufficient in finding similar solutions across starts. Age and sex covariates are then added to describe the initial and transition probabilities and the parameters are re-estimated with a warm start using the estimated parameters for the best fitting null model for

each number of states, N . A forward selection approach is then made, where a single covariate is introduced to the age/sex model, and a similar re-estimation is completed using a warm start. The corresponding Bayesian Information Criterion (BIC) value [44] is calculated after estimating the new covariate model, and the covariate is selected and added to the base model if it returns the lowest BIC value amongst all covariates which are currently not included in the model at the selection round. This process is repeated until an improvement less than 10 occurs across all covariates. This was completed for 5 models (4 to 8 latent states) and we chose the final model using the BIC as there is strong support for its usage in identifying the most suitable number of latent states [45-47]. ~~Hospitalisation was excluded from the selection process as it is strongly correlated with unrecorded medical complications, making the clinical meaning difficult to interpret.~~

In comparing models varying in size from $N = 4$ to 8 states using the forward selection, it was found that the $N = 7$ model gave the smallestlowest BIC value in comparison to both the null (no covariates used) and forward-selected model groups (Supplementary Figure 2). For the 7 state model, the following covariates were selected in the order mentioned: oxygen therapy, infection wave, chronic respiratory disease, and corticosteroids (Supplementary Figure 4). These covariates were added consecutively to the base model that included age and sex already, which resulted in the final 7 state model presented. A patient’s history of smoking as well as therapy history at the acute stage (usage of antivirals (remdesivir), monoclonal antibodies, immunomodulators and vaccination before the acute infection) were also considered in the forward selection approach but were ultimately not selected. Hospitalisation was excluded from the list of possible covariates as it is strongly correlated with unrecorded medical complications, making the clinical meaning difficult to interpret. To evaluate model performance of the final 7 state model, the chosen set of covariates was used in a 5 fold cross validation procedure with 20 multistarts, each with a cold starting parameter vector and not utilising the 2-stage approach.

Supplementary Figure 4:

Initial Submission:

Scatter plot of parameter estimates of $\bar{\rho}^{initial}$ and $\bar{\rho}^{trans}$ for 20 multistarts using all covariates. The influence of covariate is consistent across varying number of states (N). More blue lines indicates parameters with a larger likelihood value.

Revised Submission:

Forward Selection Process across Model Sizes (4 to 8 States)

Covariate	N = 4	N = 5	N = 6	N = 7	N = 8
Hospital Admission					
Age	0	0	0	0	0
Sex	0	0	0	0	0
Infection Wave	2	2	1	2	
Chronic Respiratory Disease	1	1		3	2
Oxygen Therapy	3			1	3
Corticosteroids		4	2	4	
Monoclonal Antibodies		3			1
Immunomodulators		5			
Previous Smoker					
Antivirals (Remdesivir)					
Vaccination before Acute Infection					
Model BIC	59784.15	57741.57	56670.70	56276.02	56295.32
Difference between Model BIC and N = 7 BIC (Δ BIC)	+3508.13	+1465.55	+394.68	+0.00	+19.30

Selection of Covariates across Model Sizes. The enhanced table displays which covariates were selected during the forward model selection process for model sizes $N = 4$ to $N = 8$. The integer value within the cell indicates at which selection round was the covariates chosen, with 1 being the first covariate chosen. A grey strike indicates the covariate was not chosen. Hospitalisation was excluded from the list of possible covariates, while age and sex were always included in every model. The 7 state model was found to have the lowest BIC value.

- Please comment on the compute resources that were needed for model fitting and selection. Was this done on a standard PC/laptop?

Individual model fits with a relatively low number of hidden states and covariates can be run using a standard PC in under an hour, as demonstrated by the vignette in the Github repo. However, we required our lab cluster to conduct the model selection experiments. A section has been added describing the setup.

Line 661:

Initial Submission:

...and not utilising the 2-stage approach.

-

Revised Submission:

...and not utilising the 2-stage approach.

We used a lab-internal HPC cluster for model fitting and model selection. The

cluster comprises eight CPU nodes with dual AMD EPYC 7443 (2.85 GHz) processors and six nodes with dual AMD EPYC 7F72 (3.20 GHz) processors. Five independent jobs were submitted for models with $N = 4$ to $N = 8$ latent states. Each job was allocated 24 cores and a wall time limit of 7 days. The total CPU time consumed across all jobs was approximately 1400 core-hours.

Typos:

line 126: delete one of “measured at all times points” or “recorded at each timepoint”

line 254: “the” is missing between “reduces” and “number”

lines 256: change \rightarrow changes

line 257-258: “...we uncover how...” some word seems to be missing in the remaining sentence

line 494: observation \rightarrow observations

line 514 “trans” is missing as superscript on rho

line 528: delete one of “using” or “taking”

line 646: “be” is missing between “can” and “found”

SI, equations following line 894, second line: $S_j^t \rightarrow S_j^1$

SI, line 899: The first sentence is incomplete.

SI, line 900: d \rightarrow D (patients)

SI, line 923: The superscripts t+1 and 1 need to be deleted from nu since the steady state distribution is time-independent.

SI, line 925: eigenvectors \rightarrow the first eigenvector, a \rightarrow A

Thank you for pointing these out. They have been updated now in the text.

Reviewer #3 (Remarks on code availability):

I followed the instructions in the README file to install the code and I ran the tutorial on my laptop, which went fine. I have not reproduced the results of the paper as this had required better compute resources.